# A unified internal model theory to resolve the paradox of active versus passive self-motion sensation

**Jean Laurens\*, Dora E Angelaki**

Department of Neuroscience, Baylor College of Medicine, Houston, United States

**Abstract** Brainstem and cerebellar neurons implement an internal model to accurately estimate self-motion during externally generated ('passive') movements. However, these neurons show reduced responses during self-generated ('active') movements, indicating that predicted sensory consequences of motor commands cancel sensory signals. Remarkably, the computational processes underlying sensory prediction during active motion and their relationship to internal model computations during passive movements remain unknown. We construct a Kalman filter that incorporates motor commands into a previously established model of optimal passive self-motion estimation. The simulated sensory error and feedback signals match experimentally measured neuronal responses during active and passive head and trunk rotations and translations. We conclude that a single sensory internal model can combine motor commands with vestibular and proprioceptive signals optimally. Thus, although neurons carrying sensory prediction error or feedback signals show attenuated modulation, the sensory cues and internal model are both engaged and critically important for accurate self-motion estimation during active head movements.

DOI: https://doi.org/10.7554/eLife.28074.001

**\*For correspondence:** jean.laurens@gmail.com

**Competing interests:** The authors declare that no competing interests exist.

## Introduction

For many decades, research on vestibular function has used passive motion stimuli generated by rotating chairs, motion platforms or centrifuges to characterize the responses of the vestibular motion sensors in the inner ear and the subsequent stages of neuronal processing. This research has revealed elegant computations by which the brain uses an internal model to overcome the dynamic limitations and ambiguities of the vestibular sensors (*Figure 1A*; *Mayne, 1974*; *Oman, 1982*; *Borah et al., 1988*; *Glasauer, 1992*; *Merfeld, 1995*; *Glasauer and Merfeld, 1997*; *Bos et al., 2001*; *Zupan et al., 2002*; *Laurens, 2006*; *Laurens and Droulez, 2007*; *Laurens and Droulez, 2008*; *Laurens and Angelaki, 2011*; *Karmali and Merfeld, 2012*; *Lim et al., 2017*). These computations are closely related to internal model mechanisms that underlie motor control and adaptation (*Wolpert et al., 1995*; *Körding and Wolpert, 2004*; *Todorov, 2004*; *Chen-Harris et al., 2008*; *Berniker et al., 2010*; *Berniker and Kording, 2011*; *Franklin and Wolpert, 2011*; *Saglam et al., 2011*; *2014*). Neuronal correlates of the internal model of self-motion have been identified in brainstem and cerebellum (*Angelaki et al., 2004*; *Shaikh et al., 2005*; *Yakusheva et al., 2007*, *2008*, *2013*, *Laurens et al., 2013a*, *2013b*).

In the past decade, a few research groups have also studied how brainstem and cerebellar neurons modulate during active, self-generated head movements. Strikingly, several types of neurons, well-known for responding to vestibular stimuli during passive movement, lose or reduce their sensitivity during self-generated movement (*Gdowski et al., 2000*; *Gdowski and McCrea, 1999*; *Marlinski and McCrea, 2009*; *McCrea et al., 1999*; *McCrea and Luan, 2003*; *Roy and Cullen, 2001*; *2004*; *Brooks and Cullen, 2009*; *2013*; *2014*; *Brooks et al., 2015*; *Carriot et al., 2013*). In

**eLife digest** When seated in a car, we can detect when the vehicle begins to move even with our eyes closed. Structures in the inner ear called the vestibular, or balance, organs enable us to sense our own movement. They do this by detecting head rotations, accelerations and gravity. They then pass this information on to specialized vestibular regions of the brain.

Experiments using rotating chairs and moving platforms have shown that passive movements – such as car journeys and rollercoaster rides – activate the brain's vestibular regions. But recent work has revealed that voluntary movements – in which individuals start the movement themselves – activate these regions far less than passive movements. Does this mean that the brain ignores signals from the inner ear during voluntary movements? Another possibility is that the brain predicts in advance how each movement will affect the vestibular organs in the inner ear. It then compares these predictions with the signals it receives during the movement. Only mismatches between the two activate the brain's vestibular regions.

To test this theory, Laurens and Angelaki created a mathematical model that compares predicted signals with actual signals in the way the theory proposes. The model accurately predicts the patterns of brain activity seen during both active and passive movement. This reconciles the results of previous experiments on active and passive motion. It also suggests that the brain uses similar processes to analyze vestibular signals during both types of movement.

These findings can help drive further research into how the brain uses sensory signals to refine our everyday movements. They can also help us understand how people recover from damage to the vestibular system. Most patients with vestibular injuries learn to walk again, but have difficulty walking on uneven ground. They also become disoriented by passive movement. Using the model to study how the brain adapts to loss of vestibular input could lead to new strategies to aid recovery.

DOI: https://doi.org/10.7554/eLife.28074.002

contrast, vestibular afferents respond indiscriminately for active and passive stimuli (*Cullen and Minor, 2002*; *Sadeghi et al., 2007*; *Jamali et al., 2009*). These properties resemble sensory prediction errors in other sensorimotor functions such as fish electrosensation (*Requarth and Sawtell, 2011*; *Kennedy et al., 2014*) and motor control (*Tseng et al., 2007*; *Shadmehr et al., 2010*). Yet, a consistent quantitative take-home message has been lacking. Initial experiments and reviews implicated proprioceptive switches (*Figure 1B*; *Roy and Cullen, 2004*; *Cullen et al., 2011*; *Cullen, 2012*; *Carriot et al., 2013*; *Brooks and Cullen, 2014*). More recently, elegant experiments by Brooks and colleagues (*Brooks and Cullen, 2013*; *Brooks et al., 2015*) started making the suggestion that the brain predicts how self-generated motion activates the vestibular organs and subtracts these predictions from afferent signals to generate sensory prediction errors (*Figure 1C*). However, the computational processes underlying this sensory prediction have remained unclear.

Confronting the findings of studies utilizing passive and active motion stimuli leads to a paradox, in which central vestibular neurons encode self-motion signals computed by feeding vestibular signals through an internal model during passive motion (*Figure 1A*), but during active motion, efference copies of motor commands, also transformed by an internal model (*Figure 1C*), attenuate the responses of the same neurons. Thus, a highly influential interpretation is that the elaborate internal model characterized with passive stimuli would only be useful in situations that involve unexpected (passive) movements but would be unused during normal activities, because either its input or its output (*Figure 1—figure supplement 1*) would be suppressed during active movement. Here, we propose an alternative that the internal model that processes vestibular signals (*Figure 1A*) and the internal model that generates sensory predictions during active motion (*Figure 1C*) are identical. In support of this theory, we show that the processing of motor commands must involve an internal model of the physical properties of the vestibular sensors, identical to the computations described during passive motion, otherwise accurate self-motion estimation would be severely compromised during actively generated movements.

The essence of the theory developed previously for passive movements is that the brain uses an internal representation of the laws of physics and sensory dynamics (which has been elegantly modeled as forward internal models of the sensors) to process vestibular signals. In contrast, although it

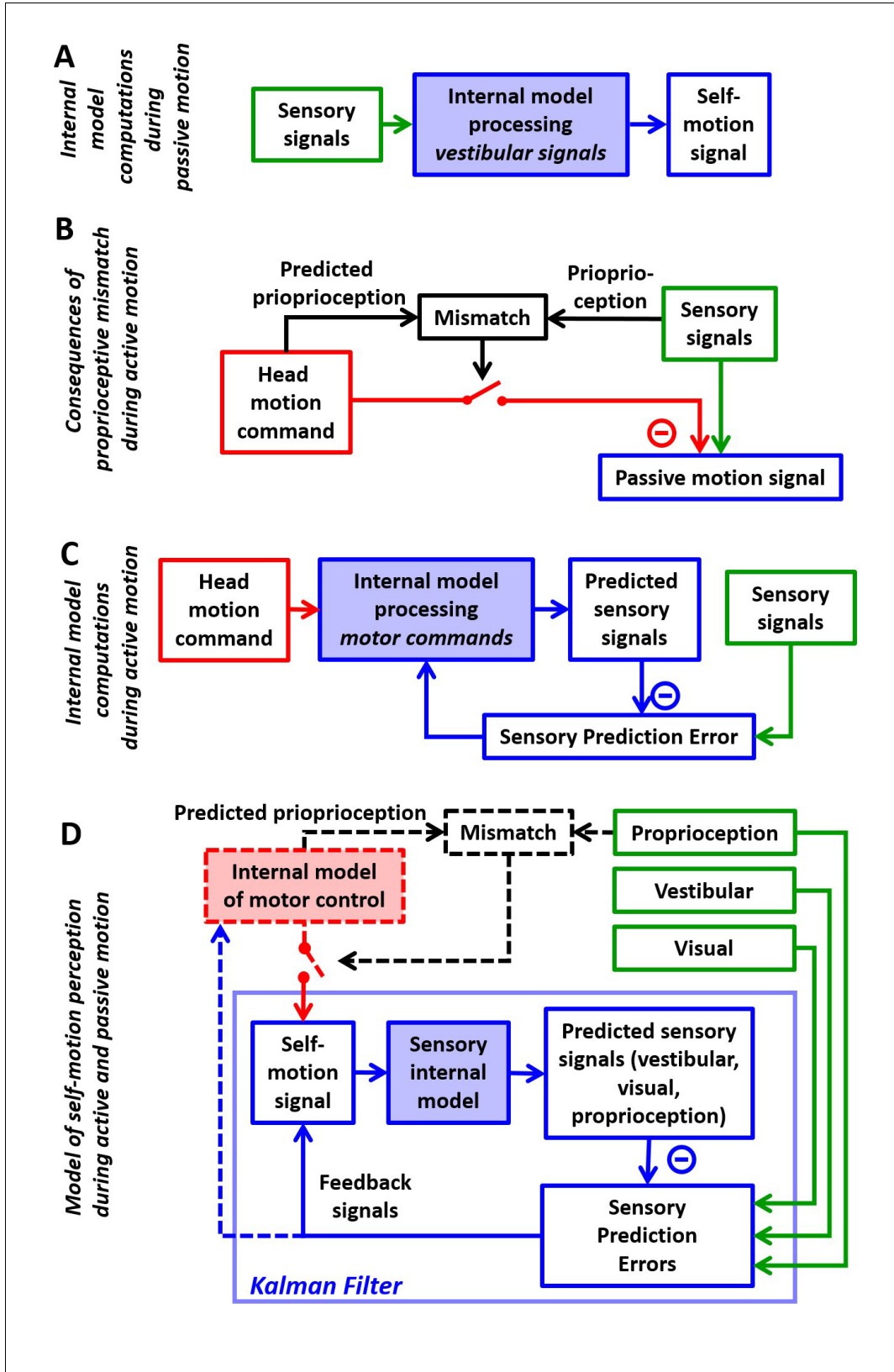

**Figure 1.** Internal model computations for self-motion estimation. (**A**) Previous studies based on passive stimuli have proposed that vestibular sensory signals are processed by an internal model to compute optimal estimates of self-motion. (**B**) Other studies (**Roy and Cullen, 2004**) have shown that the cancellation of central vestibular

*Figure 1 continued on next page*

*Figure 1 continued*

responses during active motion is gated by mismatches between predicted and actual neck proprioceptive signals, and interpreted central vestibular responses as a passive motion signal. (C) Brooks and Cullen (*Brooks et al., 2015*; *Brooks and Cullen, 2013*) have proposed that an internal model processes motor commands to compute sensory predictions during active motion and that central vestibular neurons encode sensory prediction errors. (D) Framework proposed in this study, in which the internal models in (A) and (C) are in fact identical and interactions between head motion commands and sensory signals are modeled as a Kalman filter (blue) that computes optimal self-motion estimates during both passive and active motions. For simplicity, we have not included how head motion commands are generated (red), how head movements are executed, as well as the contribution of feedback to error correction and motor learning (dashed blue arrow). In line with B, the proprioceptive gating mechanism in D is shown as a switch controlling the transmission of head motion commands to the Kalman filter. Solid lines in (D): computations modeled as a Kalman filter. Broken lines in (D): additional computations that are only discussed (but not modeled) in the present study. Note that the 'self-motion signal' box in D stands for both the predicted motion and final self-motion estimate (*Figure 1—figure supplement 2*).

DOI: https://doi.org/10.7554/eLife.28074.003

The following figure supplements are available for figure 1:

**Figure supplement 1.** Alternative computational schemes where separate internal models process vestibular signals and motor commands.

DOI: https://doi.org/10.7554/eLife.28074.004

**Figure supplement 2.** Generic structure of a Kalman Filter.

DOI: https://doi.org/10.7554/eLife.28074.005

---

is understood that transforming head motor commands into sensory predictions is likely to also involve internal models, no explicit mathematical implementation has ever been proposed for explaining the response attenuation in central vestibular areas. A survey of the many studies by Cullen and colleagues even questions the origin and function of the sensory signals canceling vestibular afferent activity, as early studies emphasized a critical role of neck proprioception in *gating* the cancellation signal (*Figure 1B*, *Roy and Cullen, 2004*), whereas follow-up studies proposed that the brain computes sensory prediction errors, without ever specifying whether the implicated forward internal models involve vestibular or proprioceptive cues (*Figure 1C*, *Brooks et al., 2015*). This lack of quantitative analysis has obscured the simple solution, which is that transforming motor commands into sensory predictions requires exactly the same forward internal model that has been used to model passive motion. We show that all previous experimental findings during both active and passive movements can be explained by a single sensory internal model that is used to generate optimal estimates of self-motion (*Figure 1D*, 'Kalman filter'). Because we focus on sensory predictions and self-motion estimation, we do not model in detail the motor control aspects of head movements and we consider the proprioception gating mechanism as a switch external to the Kalman filter, similar to previous studies (*Figure 1D*, black dashed lines and red switch).

We use the framework of the Kalman filter (*Figure 1D*; *Figure 1—figure supplement 2*; *Kalman, 1960*), which represents the simplest and most commonly used mathematical technique to implement statistically optimal dynamic estimation and explicitly computes sensory prediction errors. We build a quantitative Kalman filter that integrates motion signals originating from motor, canal, otolith, vision and neck proprioceptor signals during active and passive rotations, tilts and translations. We show how the same internal model must process both active and passive motion stimuli, and we provide quantitative simulations that reproduce a wide range of behavioral and neuronal responses, while simultaneously demonstrating that the alternative models (*Figure 1—figure supplement 1*) do not. These simulations also generate testable predictions, in particular which passive stimuli should induce sensory errors and which should not, that may motivate future studies and guide interpretation of experimental findings. Finally, we summarize these internal model computations into a schematic diagram, and we discuss how various populations of brainstem and cerebellar neurons may encode the underlying sensory error or feedback signals.

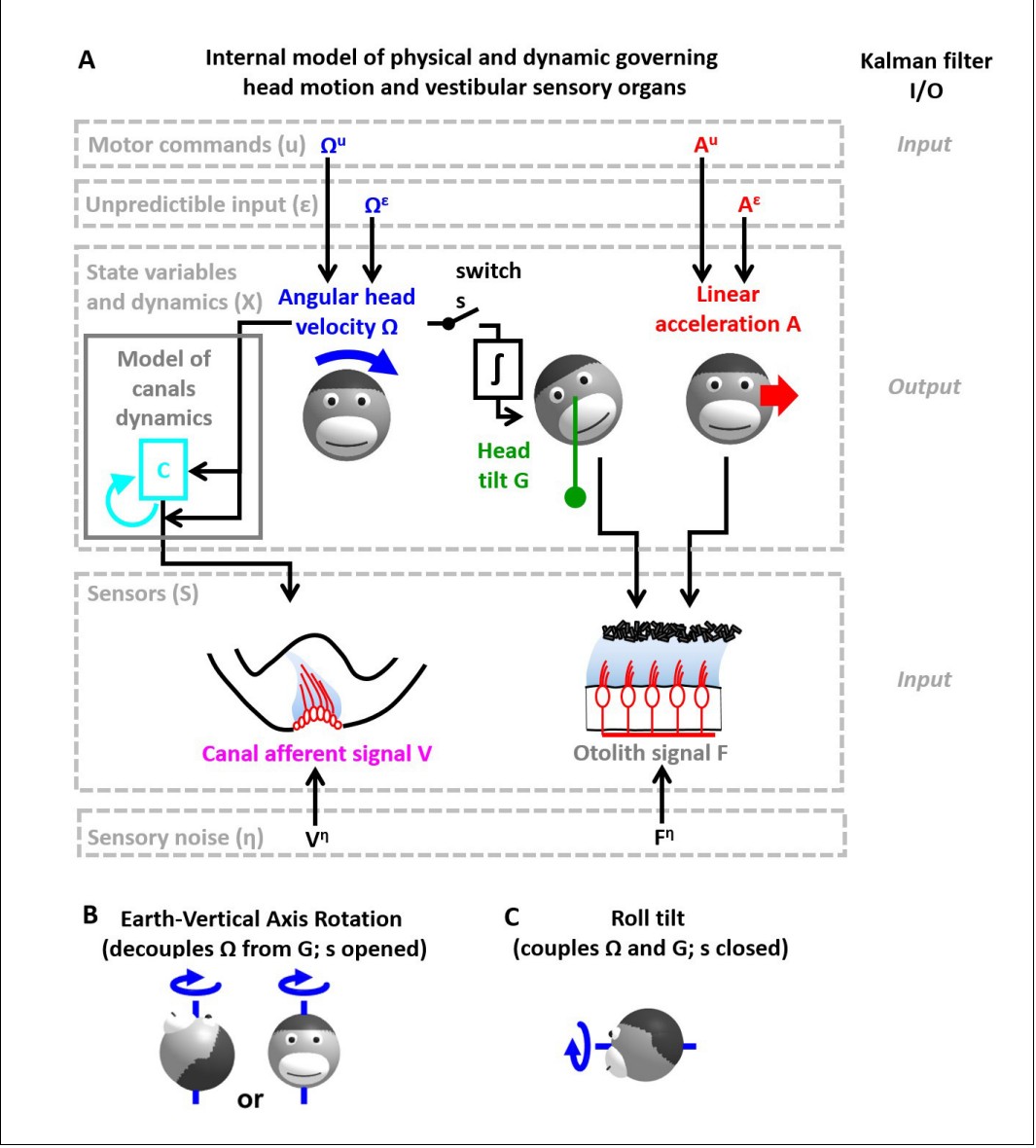

**Figure 2.** Application of the Kalman filter algorithm into optimal self-motion estimation using an internal model with four state variables and two vestibular sensors. (**A**) Schematic diagram of the model. Inputs to the model include motor commands, unexpected perturbations, as well as sensory signals. Motor commands during active movements, that is angular velocity ($\Omega^u$) and translational acceleration ($A^u$), are known by the brain. Unpredicted internal or external factors such as external (passive) motion are modeled as variables $\Omega^\varepsilon$ and $A^\varepsilon$. The state variable has 4 degrees of freedom: angular velocity $\Omega$, tilt position $G$, linear acceleration $A$ and a hidden variable $C$ used to model the dynamics of the semicircular canals (see Materials and methods). Two sensory signals are considered: semicircular canals (rotation sensors that generate a signal $V$) and the otoliths organs (linear acceleration sensors that generate a signal $F$). Sensory noise $V^\eta$ and $F^\eta$ is illustrated here but omitted from all simulations for simplicity. (**B, C**) illustration of rotations around earth-vertical (**B**) and earth-horizontal (**C**) axes.
DOI: https://doi.org/10.7554/eLife.28074.006

# Results

## Overview of Kalman filter model of head motion estimation

The structure of the Kalman filter in *Figure 1D* is shown with greater detail in *Figure 1—figure supplement 2* and described in Materials and methods. In brief, a Kalman filter (*Kalman, 1960*) is based

on a forward model of a dynamical system, defined by a set of state variables $X$ that are driven by their own dynamics, motor commands and internal or external perturbations. A set of sensors, grouped in a variable $S$, provide sensory signals that reflect a transformation of the state variables. Note that $S(t)$ may provide ambiguous or incomplete information, since some sensors may measure a mixture of state variables, and some variables may not be measured at all.

The Kalman filter uses the available information to track an optimal internal estimate of the state variable $X$. At each time $t$, the Kalman filter computes a preliminary estimate (also called a prediction, $\hat{X}^p(t)$) and a corresponding predicted sensory signal $\hat{S}^p$. In general, the resulting state estimate $\hat{X}^p$ and the predicted sensory prediction $\hat{S}^p$ may differ from the real values $X$ and $S$. These errors are reduced using sensory information, as follows (*Figure 1—figure supplement 2B*): First, the prediction $\hat{S}^p$ and the sensory input $S$ are compared to compute a sensory error $\delta S$. Second, sensory errors are transformed into a feedback $X^k = K.\delta S$, where $K$ is a matrix of feedback gains, whose dimensionality depends on both the state variable $X$ and the sensory inputs. Thus, an improved estimate at time $t$ is $\hat{X}(t) = \hat{X}^p(t) + K.\delta S(t)$. The feedback gain matrix $K$ determines how sensory errors improve the final estimate $\hat{X}$ (see Supplementary methods, 'Kalman filter algorithm' for details).

*Figure 2* applies this framework to the problem of estimating self-motion (rotation, tilt and translation) using vestibular sensors, with two types of motor commands: angular velocity ($\Omega^u$) and translational acceleration ($A^u$), with corresponding unpredicted inputs, $\Omega^\varepsilon$ and $A^\varepsilon$ (*Figure 2A*) that represent passive motion or motor error (see Discussion: 'Role of the vestibular system during active motion: fundamental, ecological and clinical implications'). The sensory signals ($S$) we consider initially encompass the semicircular canals (rotation sensors that generate a sensory signal $V$) and the otoliths organs (linear acceleration sensors that generate a sensory signal $F$) – proprioception is also added in subsequent sections. Each of these sensors has distinct properties, which can be accounted for by the internal model of the sensors. The semicircular canals exhibit high-pass dynamic properties, which are modeled by another state variable $C$ (see Supplementary methods, 'Model of head motion and vestibular sensors'). The otolith sensors exhibit negligible dynamics, but are fundamentally ambiguous: they sense gravitational as well as linear acceleration – a fundamental ambiguity resulting from Einstein's equivalence principle [*Einstein, 1907*; modeled here as $G(t) = \int \Omega(t).dt$ and $F(t) = G(t) + A(t)$; note that $G$ and $A$ are expressed in comparable units; see Materials and methods; 'Simulation parameters']. Thus, in total, the state variable $X$ has 4-degrees of freedom (*Figure 2A*): angular velocity $\Omega$ and linear acceleration $A$ (which are the input/output variables directly controlled), as well as $C$ (a hidden variable that must be included to model the dynamics of the semicircular canals) and tilt position $G$ (another hidden variable that depends on rotations $\Omega$, necessary to model the sensory ambiguity of the otolith organs).

The Kalman filter computes optimal estimates $\hat{\Omega}(t)$, $\hat{G}(t)$, $\hat{A}(t)$ and $\hat{C}(t)$ based on motor commands and sensory signals. Note that we do not introduce any tilt motor command, as tilt is assumed to be controlled only indirectly though rotation commands ($\Omega^u$). For simplicity, we restrict self-motion to a single axis of rotation (e.g. roll) and a single axis of translation (inter-aural). The model can simulate either rotations in the absence of head tilt (e.g. rotations around an earth-vertical axis: EVAR, *Figure 2B*) or tilt (*Figure 2C*, where tilt is the integral of rotation velocity, $G(t) = \int \Omega(t).dt$) using a switch (but see Supplementary methods, 'Three-dimensional Kalman filter' for a 3D model). Sensory errors are used to correct internal motion estimates using the Kalman gain matrix, such that the Kalman filter as a whole performs optimal estimation. In theory, the Kalman filter includes a total of eight feedback signals, corresponding to the combination of two sensory (canal and otolith) errors and four internal states ($\hat{\Omega}(t)$, $\hat{G}(t)$, $\hat{A}(t)$ and $\hat{C}(t)$). From those eight feedback signals, two are always negligible (Table 2; see also Supplementary methods, 'Kalman feedback gains').

We will show how this model performs optimal estimation of self-motion using motor commands and vestibular sensory signals in a series of increasingly complex simulations. We start with a very short (0.2 s) EVAR stimulus, where canal dynamics are negligible (*Figure 3*), followed by a longer EVAR that highlights the role of an internal model of the canals (*Figure 4*). Next, we consider the more complex tilt and translation movements that require all four state variables to demonstrate how canal and otolith errors interact to disambiguate otolith signals (*Figures 5* and *6*). Finally, we extend our model to simulate independent movement of the head and trunk by incorporating neck proprioceptive sensory signals (*Figure 7*). For each motion paradigm, identical active and passive motion simulations will be shown side by side in order to demonstrate how the internal model

**Table 1.** List of motion variables and mathematical notations.

| | |
|---|---|
| | *Motion variables* |
| $\Omega$ | Head rotation velocity (in space) |
| $G$ | Head Tilt |
| $A$ | Linear Acceleration |
| $C$ | Canals dynamics |
| $\Omega_{TS}$ | Trunk in space rotation velocity (variant of the model) |
| $\Omega_{HT}$ | Head on trunk rotation velocity (variant of the model) |
| $N$ | Neck position (variant of the model) |
| $X$ | Matrix containing all motion variables in a model |
| | *Sensory variables* |
| $V$ | Semicircular canal signal |
| $F$ | Otolith signal |
| $P$ | Neck proprioceptive signal |
| $Vis$ | Visual rotation signal |
| $S$ | Matrix containing all sensory variables in a model |
| | *Accent and superscripts (motion variables)* |
| $X$ | Real value of a variable |
| $\hat{X}$ | Final estimate |
| $\hat{X}^p$ | Predicted (or preliminary) estimate |
| $X^u$ | Motor command affecting the variable |
| $X^\varepsilon$ | Perturbation or motor error affecting the variable (standard deviation $\sigma_X$) |
| $\sigma_X$ | Standard deviation of $X^\varepsilon$ |
| $X^k$ | Kalman feedback on the variable |
| | *Accent and superscripts (sensory variables)* |
| $S$ | Real value of a variable |
| $\hat{S}^p$ | Predicted value |
| $S^\eta$ | Sensory noise |
| $\sigma_S$ | Standard deviation of $S^\eta$ |
| $\delta S$ | Sensory error |
| $k_{\delta S}^X$ | Kalman gain of the feedback from S to a motion variable X |
| | *Other* |
| $\delta t$ | Time step used in the simulations |
| $M'$ | Transposed of a matrix M |
| $\tau_c$ | Time constant of the semicircular canals |

DOI: https://doi.org/10.7554/eLife.28074.026

integrates sensory information and motor commands. We show that the Kalman feedback plays a preeminent role, which explains why lots of brain machinery is devoted to its implementation (see Discussion). For convenience, all mathematical notations are summarized in *Table 1*. For Kalman feedback gain nomenclature and numerical values, see *Table 2*.

## Passive motion induces sensory errors

In *Figure 3*, we simulate rotations around an earth-vertical axis (*Figure 3A*) with a short duration (0.2 s, *Figure 3B*), chosen to minimize canal dynamics ($C \approx 0$, *Figure 3B*, cyan) such that the canal response matches the velocity stimulus ($V \approx \Omega$, compare magenta curve in *Figure 3C* with blue curve in *Figure 3B*). We simulate active motion (*Figure 3D–K*, left panels), where $\Omega = \Omega^u$ (*Figure 3D*) and

**Table 2.** Kalman feedback gains during EVAR and tilt/translation.

Some feedback gains are constant independently of $\delta t$ while some other scale with $\delta t$ (see Supplementary methods, 'Feedback gains' for explanations). Gains that have negligible impact on the motion estimates are indicated in normal fonts, others with profound influence are indicated in bold. The feedback gains transform error signals into feedback signals.

|  |  | Gains during EVAR | Gains during tilt | Notes |
|---|---|---|---|---|
| Canal feedbacks | $k_{\delta V}^{\Omega}$ | **0.94** | **0.94** |  |
|  | $k_{\delta V}^{C}$ | **0.19 $\delta t$** | **0.23 $\delta t$** | Integrated over time |
|  | $k_{\delta V}^{G}$ | 0.00 | **0.90 $\delta t$** | Integrated over time |
|  | $k_{\delta V}^{A}$ | 0.00 | -0.90 $\delta t$ | Negligible |
| Otolith feedbacks | $k_{\delta F}^{\Omega}$ | - | 0.00 | Negligible |
|  | $k_{\delta F}^{C}$ | - | **0.14 $\delta t$** | Integrated over time |
|  | $k_{\delta F}^{G}$ | - | **0.76 $\delta t$** | Integrated over time |
|  | $k_{\delta F}^{A}$ | - | **0.99** |  |

DOI: https://doi.org/10.7554/eLife.28074.027

$\Omega^{\varepsilon} = 0$ (not shown), as well as passive motion (**Figure 3D–K**, right panels), where $\Omega = \Omega^{\varepsilon}$ (**Figure 3D**) and $\Omega^{u} = 0$ (not shown). The rotation velocity stimulus ($\Omega$, **Figure 3E**, blue) and canal activation ($V$, **Figure 3F**, magenta) are identical in both active and passive stimulus conditions. As expected, the final velocity estimate $\hat{\Omega}$ (output of the filter, **Figure 3G**, blue) is equal to the stimulus $\Omega$ (**Figure 3E**, blue) during both passive and active conditions. Thus, this first simulation is meant to emphasize differences in the flow of information *within* the Kalman filter, rather than differences in performance between passive and active motions (which is identical).

The fundamental difference between active and passive motions resides in the prediction of head motion (**Figure 3H**) and sensory canal signals (**Figure 3I**). During active motion, the motor command $\Omega^{u}$ (**Figure 3D**) is converted into a predicted rotation $\hat{\Omega}^{p} = \Omega^{u}$ (**Figure 3H**) by the internal model, and in turn in a predicted canal signal $\hat{V}^{p}$ (**Figure 3I**). Of course, in this case, we have purposely chosen the rotation stimulus to be so short (0.2 s), such that canal afferents reliably encode the rotation

**Table 3.** Kalman feedback gains during head and neck rotation.

As in **Table 2**, some feedback gains are constant and independently of $\delta t$, while some others scale with or inversely to $\delta t$ (see Supplementary methods, 'Feedback gains of the model of head and neck motion' for explanations). Gains that have negligible impact on the motion estimates are indicated in normal fonts, others with profound influence are indicated in bold. The feedback gains $k_{\delta V}^{\Omega}$ and $k_{\delta P}^{\Omega}$ are computed as $k_{\delta V}^{\Omega} = k_{\delta V}^{\Omega_{TS}} + k_{\delta V}^{\Omega_{HT}}$ and $k_{\delta P}^{\Omega} = k_{\delta P}^{\Omega_{TS}} + k_{\delta P}^{\Omega_{HT}}$.

|  |  | Gains | Notes |
|---|---|---|---|
| Canal feedbacks | $k_{\delta V}^{\Omega TS}$ | **0.85** |  |
|  | $k_{\delta V}^{\Omega HT}$ | **0.10** |  |
|  | $k_{\delta V}^{N}$ | 0.05 $\delta t$ | Negligible |
|  | $k_{\delta V}^{C}$ | **0.22 $\delta t$** | Integrated over time |
|  | $k_{\delta V}^{\Omega}$ | **0.95** |  |
| Proprioceptive feedbacks | $k_{\delta P}^{\Omega TS}$ | **-0.84/$\delta t$** |  |
|  | $k_{\delta P}^{\Omega HT}$ | **0.89/$\delta t$** |  |
|  | $k_{\delta P}^{N}$ | **0.94** |  |
|  | $k_{\delta P}^{C}$ | 0.03 | Negligible |
|  | $k_{\delta P}^{\Omega}$ | 0.05/$\delta t$ | Negligible |

DOI: https://doi.org/10.7554/eLife.28074.028

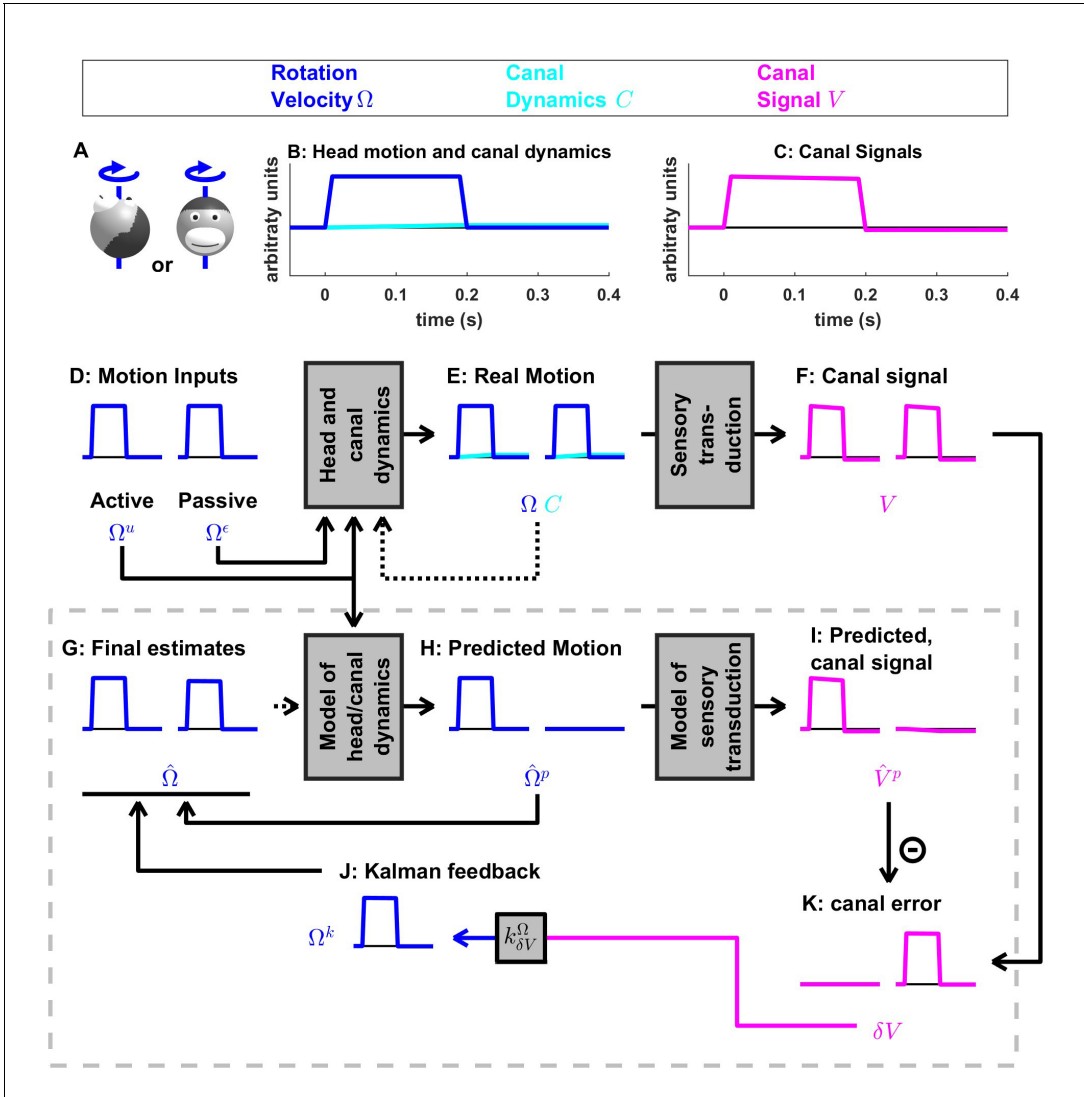

**Figure 3.** Short duration rotation around an earth-vertical axis (as in *Figure 2B*). (**A**) Illustration of the stimulus lasting, 200 ms. (**B,C**) Time course of motion variables and sensory (canal) signals. (**D–K**) Simulated variables during active (left panels) and passive motion (right panels). Only the angular velocity state variable $\Omega$ is shown (tilt position $G$ and linear acceleration $A$ are not considered in this simulation, and the hidden variable $C$ is equal to zero). Continuous arrows represent the flow of information during one time step, and broken arrows the transfer of information from one time step to the next. (**J**) Kalman feedback. For clarity, the Kalman feedback is shown during passive motion only (it is always zero during active movements in the absence of any perturbation and noise). The box defined by dashed gray lines illustrates the Kalman filter computations. For the rest of mathematical notations, see *Table 1*.

DOI: https://doi.org/10.7554/eLife.28074.007

stimulus ($V \approx \Omega$; compare *Figure 3F and E*, left panels) and the internal model of canals dynamics have a negligible contribution; that is, $\hat{\Omega}^p \approx \hat{V}^p$ (compare *Figure 3I and H*, left panels). Because the canal sensory error is null, that is $\delta V = V - \hat{V}^p \approx 0$ (*Figure 3K*, left panel), the Kalman feedback pathway remains silent (not shown) and the net motion estimate is unchanged compared to the prediction, that is, $\hat{\Omega} = \hat{\Omega}^p = \Omega^u = \Omega$. In conclusion, during active rotation (and in the absence of perturbations, motor or sensory noise), motion estimates are generated entirely based on an accurate predictive process, in turn leading to an accurate prediction of canal afferent signals. In the absence of sensory mismatch, these estimates don't require any further adjustment.

In contrast, during passive motion the predicted rotation is null ($\hat{\Omega}^p = 0$, *Figure 3H*, right panel), and therefore the predicted canal signal is also null ($\hat{V}^p = 0$, *Figure 3I*, right panel). Therefore, canal

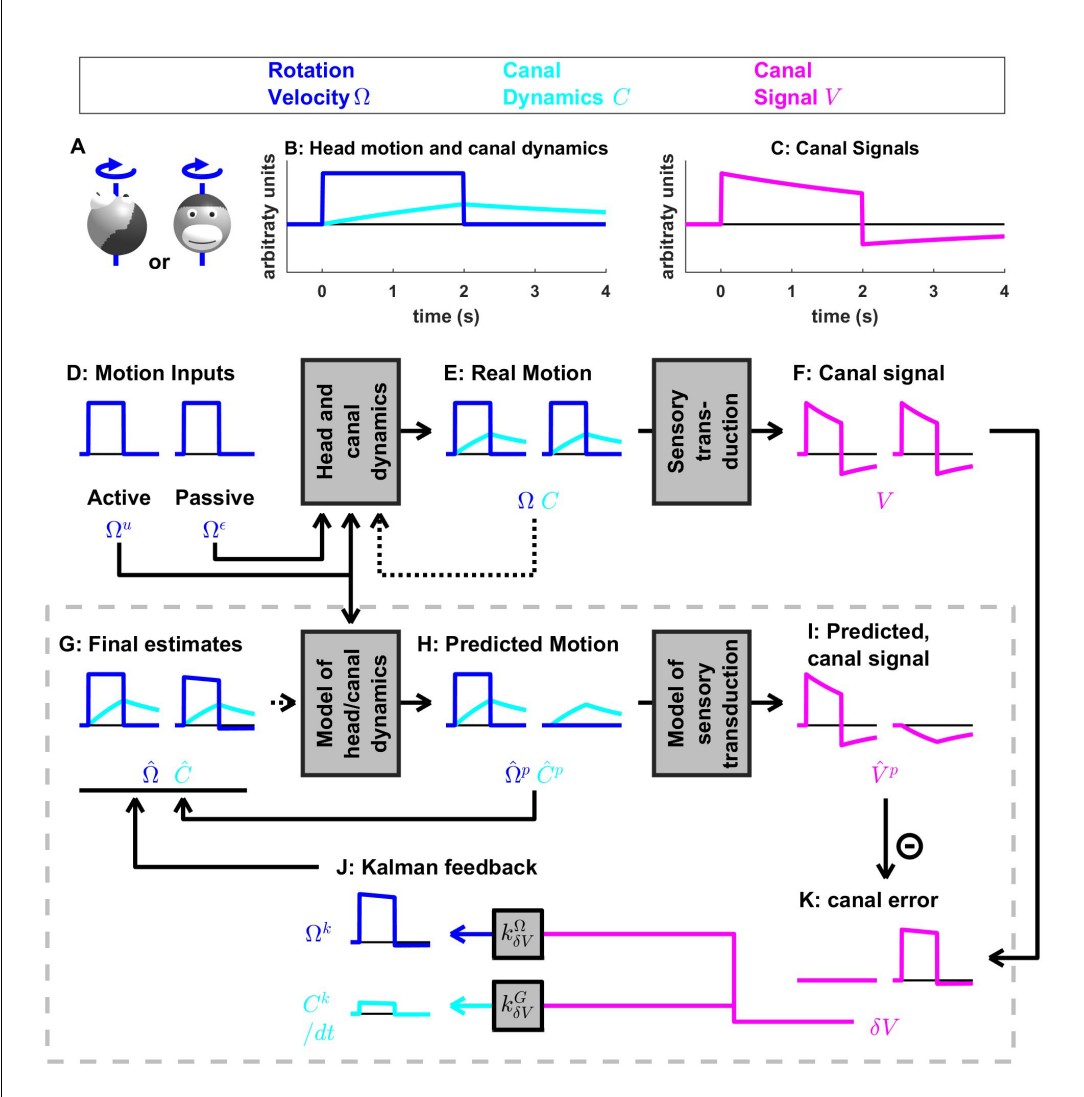

**Figure 4.** Medium-duration rotation around an earth-vertical axis, demonstrating the role of the internal model of canal dynamics. (**A**) Illustration of the stimulus lasting 2 s. (**B,C**) Time course of motion variables and sensory (canal) signals. (**D–K**) Simulated variables during active (left panels) and passive motions (right panels). Two state variables are shown: the angular velocity $\Omega$ (blue) and canal dynamics $C$ (cyan). Continuous arrows represent the flow of information during one time step, and broken arrows the transfer of information from one time step to the next. (**J**) Kalman feedback. For clarity, the Kalman feedback (reflecting feedback from the canal error signal to the two state variables) is shown during passive motion only (it is always zero during active movements in the absence of any perturbation and noise). All simulations use a canal time constant of 4 s. Note that, because of the integration, the illustrated feedback $C^k$ is scaled by a factor $1/\delta t$; see Supplementary methods, 'Kalman feedback gains'. The box defined by dashed gray lines illustrates the Kalman filter computations. For the rest of mathematical notations, see *Table 1*.

DOI: https://doi.org/10.7554/eLife.28074.008

The following figure supplements are available for figure 4:

**Figure supplement 1.** Processing of rotation information during long-duration motion.

DOI: https://doi.org/10.7554/eLife.28074.009

**Figure supplement 2.** Same simulation as in *Figure 4*, where the internal model of canals dynamic is not used.

DOI: https://doi.org/10.7554/eLife.28074.010

**Figure supplement 3.** Quantitative analysis of the importance of an internal model of the canals for ecological movements.

DOI: https://doi.org/10.7554/eLife.28074.011

signals during passive motion generate a sensory error $\delta V = V - \hat{V}^p = V$ (*Figure 3K*, right panel). This sensory error is converted into a feedback signal $\Omega^k = k_{\delta V}^{\Omega}.\delta V$ (*Figure 3J*) with a Kalman gain $k_{\delta V}^{\Omega}$ (feedback from canal error $\delta V$ to angular velocity estimate $\Omega$) that is close to 1 (*Table 2*; note

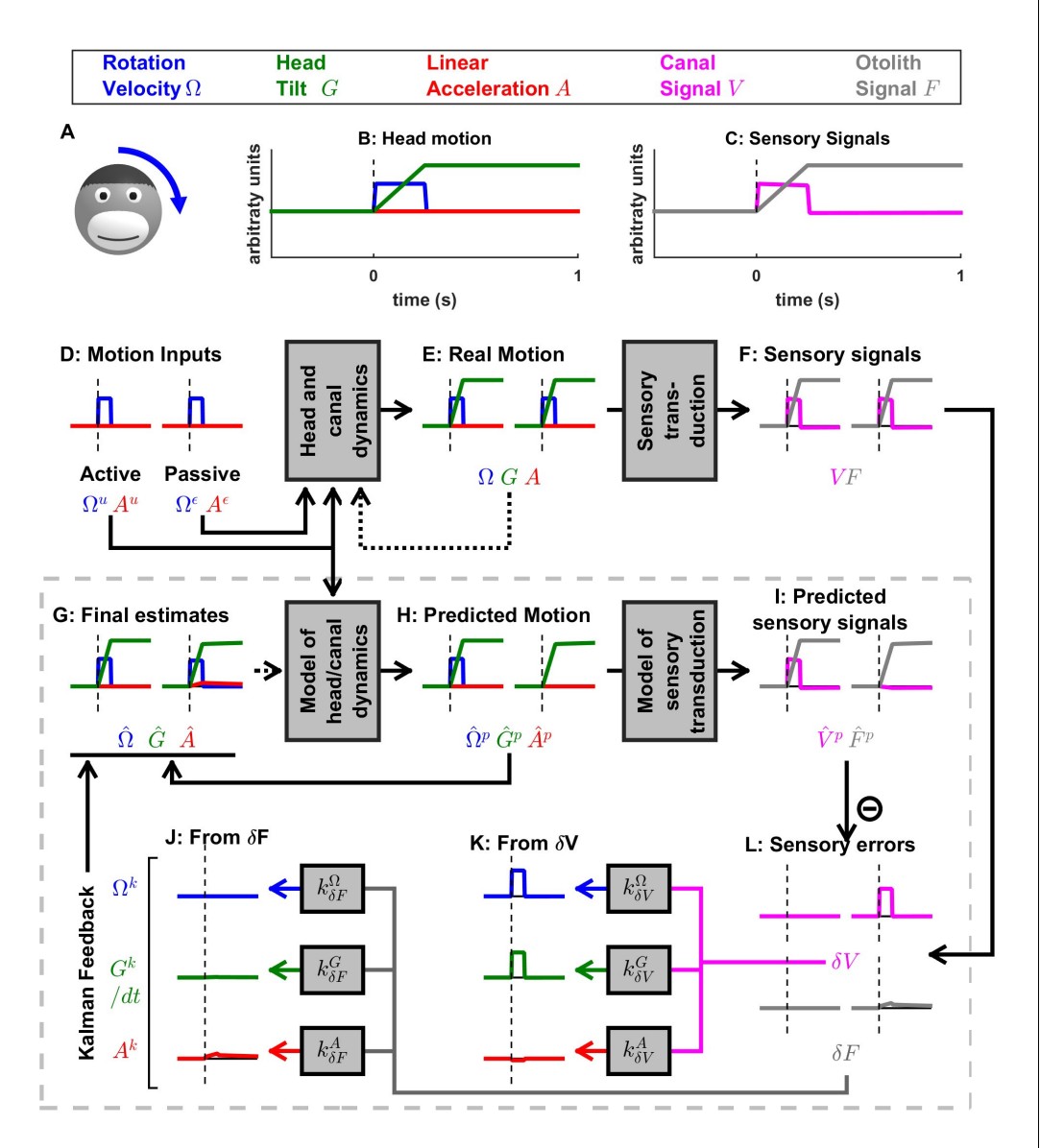

**Figure 5.** Simulation of short duration head tilt. (A) Illustration of the stimulus lasting 0.2 s. (B,C) Time course of motion variables and sensory (canal and otolith) signals. (D–L) Simulated variables during active (left panels) and passive motions (right panels). Three state variables are shown: the angular velocity $\Omega$ (blue), tilt position $G$, and linear acceleration $A$. Continuous arrows represent the flow of information during one time step, and broken arrows represent the transfer of information from one time step to the next. (J, K) Kalman feedback (shown during passive motion only). Two error signals ($\delta V$: canal error; $\delta F$: otolith error) are transformed into feedback to state variables $\Omega^k$: blue, $G^k$: green, $A^k$: red (variable $C^k$ is not shown, but see **Figure 5— figure supplement 1** for simulations of a 2-s tilt). Feedback originating from $\delta F$ is shown in (J) and from $\delta V$ in (K). The feedback to $G^k$ is scaled by a factor $1/\delta t$ (see Supplementary methods, 'Kalman feedback gains'). Note that in this simulation we consider an active ($\Omega^u$) or passive ($\Omega^\varepsilon$) rotation velocity as input. The tilt itself is a consequence of the rotation, and not an independent input. The box defined by dashed gray lines illustrates the Kalman filter computations. For the rest of mathematical notations, see **Table 1**.

DOI: https://doi.org/10.7554/eLife.28074.012

The following figure supplement is available for figure 5:

**Figure supplement 1.** Simulation of medium duration (2 s) head tilt movement.

DOI: https://doi.org/10.7554/eLife.28074.013

that this value represents an optimum and is computed by the Kalman filter algorithm). The final motion estimate is generated by this feedback, that is $\hat{\Omega} = k_{\delta V}^{\Omega}.\delta V = V \approx \Omega$.

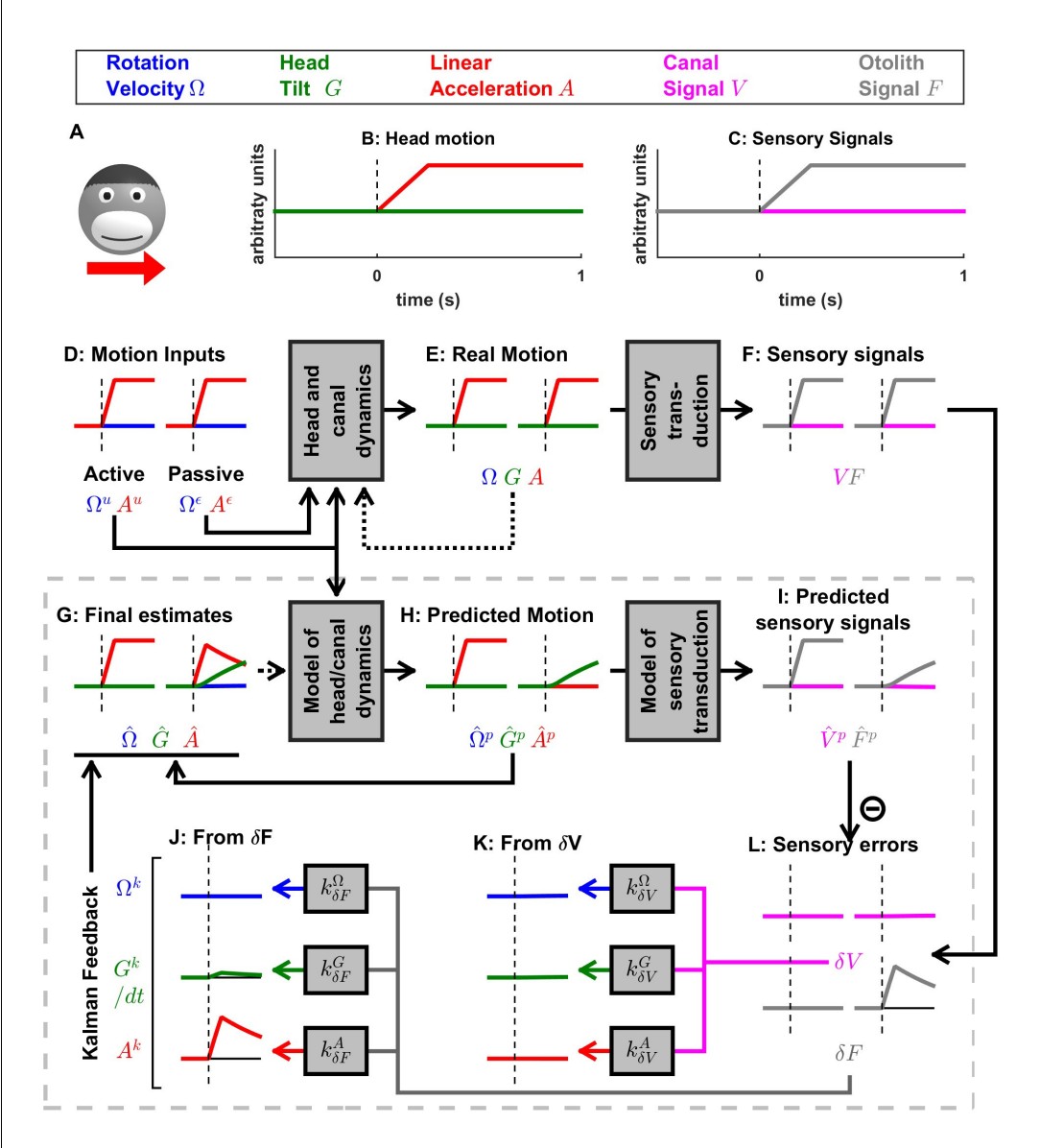

**Figure 6.** Simulation of short duration translation. Same legend as *Figure 5*. Note that *F* is identical in *Figures 5* and *6*: in terms of sensory inputs, these simulation differ only in the canal signal.

DOI: https://doi.org/10.7554/eLife.28074.014

The following figure supplements are available for figure 6:

**Figure supplement 1.** Long duration translation, demonstrating the time course of the somatogravic effect.
DOI: https://doi.org/10.7554/eLife.28074.015
**Figure supplement 2.** Simulation of simultaneous tilt and translation.
DOI: https://doi.org/10.7554/eLife.28074.016
**Figure supplement 3.** Otolith influence on rotation estimate during passive rotations.
DOI: https://doi.org/10.7554/eLife.28074.017

These results illustrate the fundamental rules of how active and passive motion signals are processed by the Kalman filter (and, as hypothesized, the brain). During active movements, motion estimates are generated by a *predictive mechanism,* where motor commands are fed into an internal model of head motion. During passive movement, motion estimates are formed based on *feedback signals* that are themselves driven by sensory canal signals. In both cases, specific nodes in the

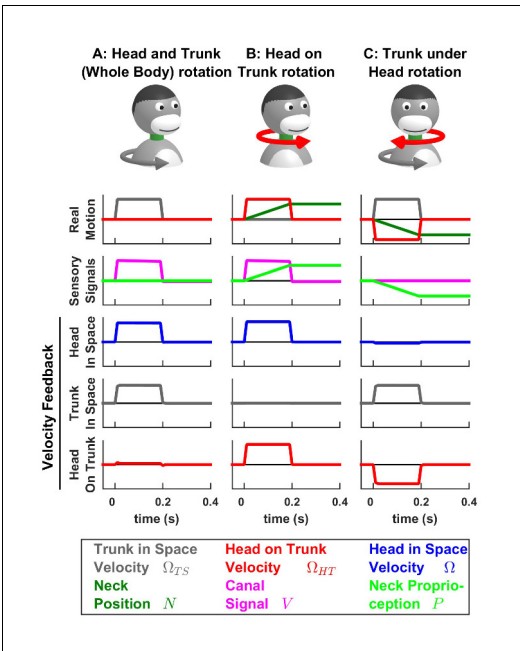

**Figure 7.** Simulations of passive trunk and head movements. We use a variant of the Kalman filter model (see Supplementary methods) that tracks the velocity of both head and trunk (trunk in space: gray; head in space: blue; head on trunk: red) based on semicircular canal and neck proprioception signals. The real motion (first line), sensory signals (second line) and velocity feedback signals (third to fifth lines) are shown during (**A**) passive whole head and trunk rotation, (**B**) passive head on trunk rotation, and (**C**) passive trunk under head rotation. See *Figure 7—figure supplement 1–3* for other variables and simulations of active motion.

DOI: https://doi.org/10.7554/eLife.28074.018

The following figure supplements are available for figure 7:

**Figure supplement 1.** Active and passive head and trunk rotation.

DOI: https://doi.org/10.7554/eLife.28074.019

**Figure supplement 2.** Active and passive head on trunk rotation.

DOI: https://doi.org/10.7554/eLife.28074.020

**Figure supplement 3.** Active and passive rotation of the trunk while the head is stationary.

DOI: https://doi.org/10.7554/eLife.28074.021

**Figure supplement 4.** Simulated tuning of unimodal and bimodal neuron as a function of neck position offset.

DOI: https://doi.org/10.7554/eLife.28074.022

**Figure supplement 5.** Long-duration passive head and trunk movements.

DOI: https://doi.org/10.7554/eLife.28074.023

**Figure supplement 6.** Modeling the impact of perturbing motor activity during active movement.

DOI: https://doi.org/10.7554/eLife.28074.024

*Figure 7 continued on next page*

network are silent (e.g. predicted canal signal during passive motion, *Figure 3I*; canal error signal during active motion, *Figure 3K*), but the same network operates in unison under *all* stimulus conditions. Thus, depending on whether the neuron recorded by a microelectrode in the brain carries *predicted*, *actual* or *error* sensory signals, differences in neural response modulation are expected between active and passive head motion. For example, if a cell encodes canal error exclusively, it will show maximal modulation during passive rotation, and no modulation at all during active head rotation. If a cell encodes mixtures of canal sensory error and actual canal sensory signals (e.g. through a direct canal afferent input), then there will be non-zero, but attenuated, modulation during active, compared to passive, head rotation. Indeed, a range of response attenuation has been reported in the vestibular nuclei (see Discussion).

We emphasize that in *Figure 3* we chose a *very* short-duration (0.2 s) motion profile, for which semicircular canal dynamics are negligible and the sensor can accurately follow the rotation velocity stimulus. We now consider more realistic rotation durations, and demonstrate how predictive and feedback mechanisms interact for accurate self-motion estimation. Specifically, canal afferent signals attenuate (because of their dynamics) during longer duration rotations – and this attenuation is already sizable for rotations lasting 1 s or longer. We next demonstrate that the internal model of canal dynamics must be engaged for accurate rotation estimation, even during purely actively generated head movements.

## Internal model of canals

We now simulate a longer head rotation, lasting 2 s (*Figure 4A,B*, blue). The difference between the actual head velocity $\Omega$ and the average canal signal $V$ is modeled as an internal state variable $C$, which follows low-pass dynamics (see Supplementary methods, 'Model of head motion and vestibular sensors'). At the end of the 2 s rotation, the value of $C$ reaches its peak at ~40% of the rotation velocity (*Figure 4B*, cyan), modeled to match precisely the afferent canal signal $V$, which decreases by a corresponding amount (*Figure 4C*). Note that $C$ persists when the rotation stops, matching the canal aftereffect ($V = -C < 0$ after t > 2 s). Next, we demonstrate how the Kalman filter uses the internal variable $C$ to compensate for canal dynamics.

During active motion, the motor command $\Omega^u$ (*Figure 4D*) is converted into an accurate

*Figure 7 continued*

**Figure supplement 7.** Neuronal responses when perturbing motor activity during active movement.
DOI: https://doi.org/10.7554/eLife.28074.025

prediction of head velocity $\hat{\Omega}^p$ (*Figure 4H*, blue). Furthermore, $\Omega^u$ is also fed through the internal model of the canals to predict $\hat{C}^p$ (*Figure 4H*, cyan). By combining the predicted internal state variables $\hat{\Omega}^p$ and $\hat{C}^p$, the Kalman filter computes a canal prediction $\hat{V}^p$ that fol-

lows the same dynamics as $V$ (compare *Figure 4F and I*, left panels). Therefore, as in *Figure 3*, the resulting sensory mismatch is $\delta V = V - \hat{V}^p \approx 0$ and the final estimates (*Figure 4G*) are identical to the predicted estimates (*Figure 4H*). Thus, the Kalman filter maintains an accurate rotation estimate by feeding motor commands through an internal model of the canal dynamics. Note, however, that because in this case $V \neq \Omega$ (compare magenta curve in *Figure 4F* and blue curve in *Figure 4E*, left panels), $\hat{V}^p \neq \hat{\Omega}^p$ (compare magenta curve in *Figure 4I* and blue curve in *Figure 4H*, left panels). Thus, the sensory mismatch can only be null under the assumption that motor commands have been processed through the internal model of the canals. But before we elaborate on this conclusion, let's first consider passive stimulus processing.

During passive motion, the motor command $\Omega^u$ is equal to zero. First, note that the final estimate $\hat{\Omega} \approx \Omega$ is accurate (*Figure 4G*), as in *Figure 3G*, although canal afferent signals don't encode $\Omega$ accurately. Second, note that the internal estimate of canal dynamics $\hat{C}$ (*Figure 4G*) and the corresponding prediction ($\hat{C}^p$; *Figure 4H*) are both accurate (compare with *Figure 4E*). This occurs because the canal error $\delta V$ (*Figure 4K*) is converted into a second feedback, $C^k$, (*Figure 4J*, cyan), which updates the internal estimate $\hat{C}$ (see Supplementary methods, 'Velocity Storage'). Finally, in contrast to *Figure 3*, the canal sensory error $\delta V$ (*Figure 4K*) does not follow the same dynamics as $V$ (*Figure 4C,F*), but is (as it should) equal to $\Omega$ (*Figure 4B*). This happens because, though a series of steps ($\hat{V}^p = -\hat{C}^p$

in *Figure 4I* and $\delta V = V - \hat{V}^p$ in *Figure 4K*), $\hat{C}^p$ is added to the vestibular signal $V$ to compute $\delta V \approx \Omega$. This leads to the final estimate $\hat{\Omega} = \hat{\Omega}^p = \delta V \approx \Omega$ (*Figure 4G*). Model simulations during even longer duration rotations and visual-vestibular interactions are illustrated in *Figure 4—figure supplement 1*. Thus, the internal model of canal dynamics improves the rotation estimate during passive motion. Remarkably, this is important not only during very long duration rotations (as is often erroneously presumed), but also during short stimuli lasting 1–2 s, as illustrated with the simulations in *Figure 4*.

We now return to the actively generated head rotations to ask the important question: What would happen if the brain didn't use an internal model of canal dynamics? We simulated motion estimation where canal dynamics were removed from the internal model used by the Kalman filter (*Figure 4—figure supplement 2*). During both active and passive motion, the net estimate $\hat{\Omega}$ is inaccurate as it parallels $V$, exhibiting a decrease over time and an aftereffect. In particular, during active motion, the motor commands provide accurate signals $\hat{\Omega}^p$, but the internal model of the canals fails to convert them into a correct prediction $\hat{V}^p$, resulting in a sensory mismatch. This mismatch is converted into a feedback signal $\Omega^k$ that degrades the accurate prediction $\hat{\Omega}^p$ such that the final estimate $\hat{\Omega}$ is inaccurate. These

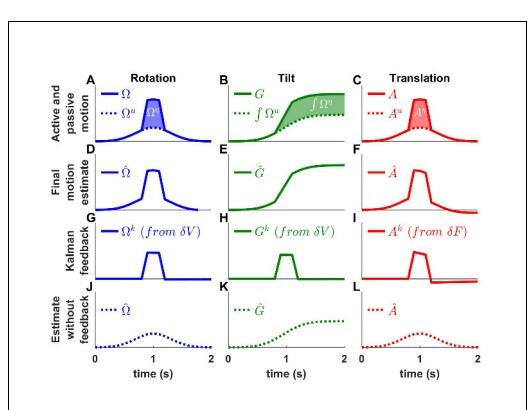

**Figure 8.** Interaction of active and passive motion. Active movements (Gaussian profiles) and passive movements (short trapezoidal profiles) are superimposed. (**A**) Active ($\Omega^u$) and passive ($\Omega^\varepsilon$) rotations. (**B**) Head tilt resulting from active and passive rotations (the corresponding tilt components are $\int \Omega^u.dt$ and $\int \Omega^\varepsilon.dt$). (**C**) Active ($A^u$) and passive ($A^\varepsilon$) translations. (**D-F**) Final motion estimates (equal to the total motion). (**G-I**) The Kalman feedback corresponds to the passive motion component. (**J-K**) Final estimates computed by inactivating all Kalman feedback pathways. These simulations represent the motion estimates that would be produced if the brain suppressed sensory inflow during active motion. The simulations contradict the alternative scheme of *Figure 1—figure supplement 1A*.
DOI: https://doi.org/10.7554/eLife.28074.029

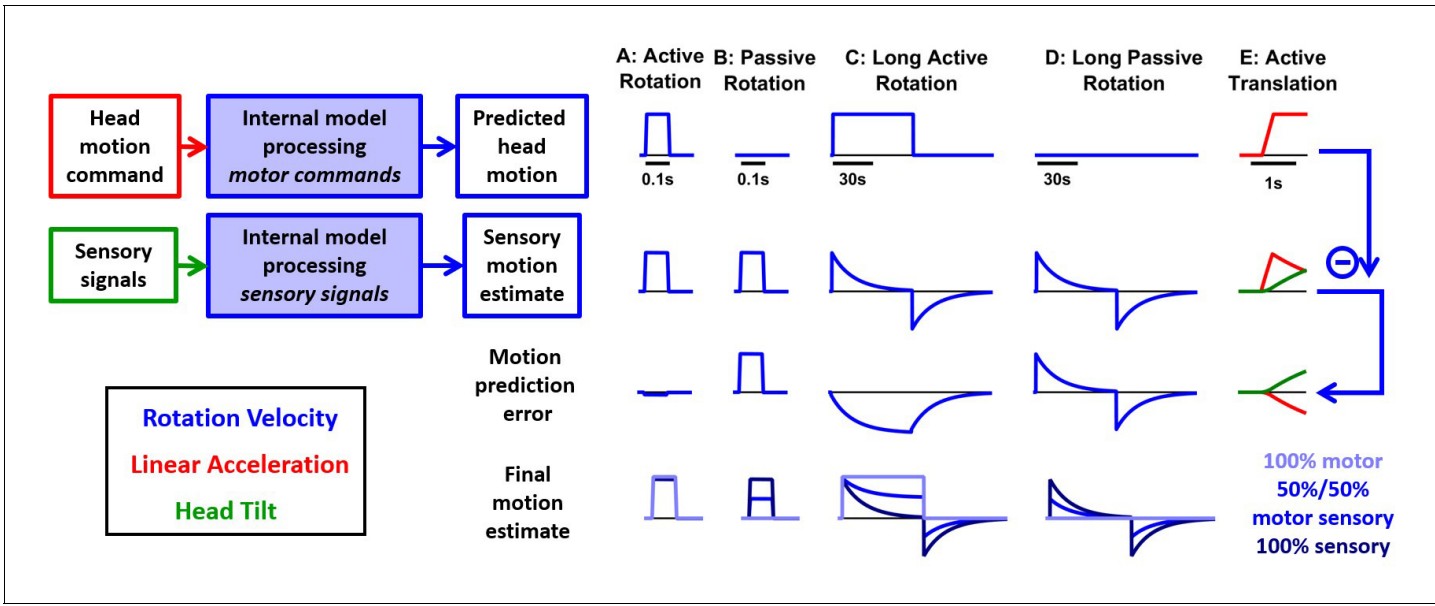

**Figure 9.** Simulations of the alternative scheme where motor commands cancel the output of a sensory internal model. In this figure, we consider an alternative scheme (*Figure 1—figure supplement 1B*), where the motor commands (first row), which are assumed to encode head angular velocity and linear acceleration (as in the Kalman filer model), are used to cancel the output of a 'sensory only' internal model (second row, also identical to the Kalman filter model) to compute motion prediction errors (third row), instead of sensory prediction error, as in the Kalman filter model. (A) During a short active rotation (same as in *Figure 3*), both the motor prediction and the sensory self-motion estimate are close to the real motion and therefore the motor prediction cancels the sensory estimate accurately. (B) Similarly to *Figure 3*, the motor prediction is null and the sensory estimate is not cancelled during passive rotation. (C) During long-duration active rotation (same motion as in *Figure 4—figure supplement 1A,B*), the motor prediction (top row) does not match the sensory signal (second row), resulting in a substantial prediction error (third row). This contrasts with Kalman filter simulations, where no sensory prediction errors occur during active motion. (D) During long-duration passive motion, results agree with Kalman filter predictions. (E) During active translation (same motion as in *Figure 6*), the somatogravic effect would induce a tilt illusion (green) and an underestimation of linear acceleration (red), again leading to motion prediction errors. Thus, although the predictions of this alternative model resemble those of the Kalman filter during short active rotations, they differ during long rotations or active translations and are contradicted by experimental observations. The last row shows the final self-motion estimate obtained by computing a weighted average of the predicted head motion and sensory motion estimates. Three different weights are considered: 100% motor signals (dark blue), 100% sensory estimates (dark blue) or 50%/50% (blue).

DOI: https://doi.org/10.7554/eLife.28074.030

simulations highlight the role of the internal model of canal dynamics, which continuously integrates rotation information in order to anticipate canal afferent activity during *both* active and passive movements. Without this sensory internal model, active movements would result in sensory mismatch, and the brain could either transform this mismatch into sensory feedback, resulting in inaccurate motion estimates, or ignore it and lose the ability to detect externally generated motion or movement errors. Note that the impact of canal dynamics is significant even during natural short-duration and high-velocity head rotations (*Figure 4—figure supplement 3*). Thus, even though particular nodes (neurons) in the circuit (e.g. vestibular and rostral fastigial nuclei cells presumably reflecting either $\delta V$ or $\Omega^k$ in *Figures 3* and *4*; see Discussion) are attenuated or silent during active head rotations, efference copies of motor commands must *always* be processed though the internal model of the canals – motor commands *cannot* directly drive appropriate sensory prediction errors. This intuition has remained largely unappreciated by studies comparing how central neurons modulate during active and passive rotations – a misunderstanding that has led to a fictitious dichotomy belittling important insights gained by decades of studies using passive motion stimuli (see Discussion).

## Active versus passive tilt

Next, we study the interactions between rotation, tilt and translation perception. We first simulate a short duration (0.2 s) roll tilt (*Figure 5A*; with a positive tilt velocity $\Omega$, *Figure 5B*, blue). Tilt position ($G$,

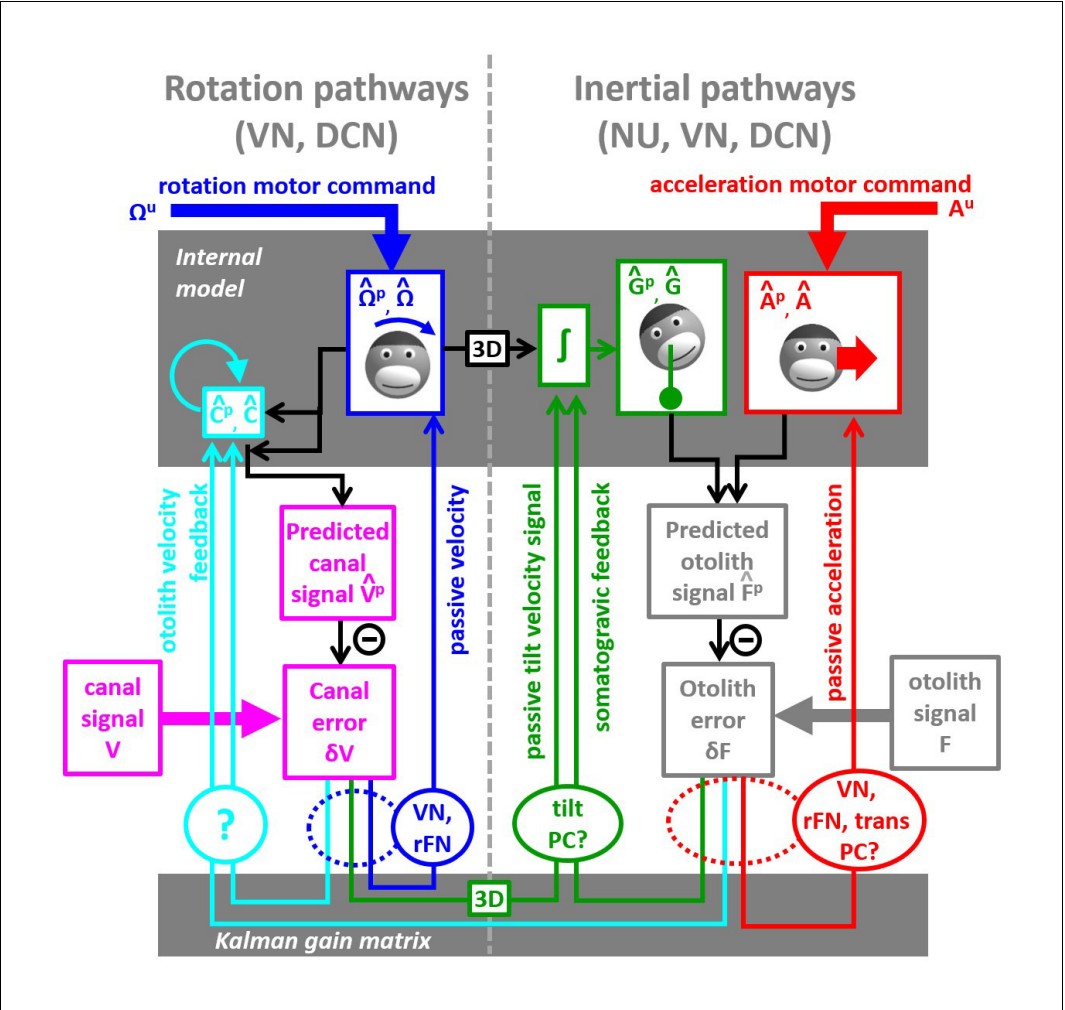

**Figure 10.** Schematic diagram of central vestibular computations. This diagram is organized to offer a synthetic view of the processing elements, as well as their putative neural correlates. An internal model (top gray box) predicts head motion based on motor commands and receives feedback signals. The internal model computes predicted canal and otolith signals that are compared to actual canal and otolith inputs. The resulting sensory errors are transformed by the Kalman gain matrix into a series of feedback 'error' signals. Left: canal error feedback signals; Right: otolith error feedback signals. Rotation signals are spatially transformed ('3D' boxes) into tilt velocity signals. Ovals indicate putative neuronal correlates of the feedback signals (VN: vestibular only vestibular nuclei neurons; rFN: rostral fastigial nuclei neurons, PC: Purkinje cells in the caudal vermis, DCN: deep cerebellar nuclei).

DOI: https://doi.org/10.7554/eLife.28074.031

The following figure supplements are available for figure 10:

**Figure supplement 1.** Alternative diagram.
DOI: https://doi.org/10.7554/eLife.28074.032

**Figure supplement 2.** Influence of model parameters.
DOI: https://doi.org/10.7554/eLife.28074.033

*Figure 5B*, green) ramps during the rotation and then remains constant. As in *Figure 3*, canal dynamics $C$ are negligible ($V \approx \Omega$; *Figure 5F*, magenta) and the final rotation estimate $\hat{\Omega}$ is accurate (*Figure 5G*, blue). Also similar to *Figure 3*, $\hat{\Omega}$ is carried by the predicted head velocity node during active motion ($\hat{\Omega} \approx \hat{\Omega}^p$; $\Omega^k \approx 0$) and by the Kalman feedback node during passive motion ($\Omega \approx \Omega^k$; $\hat{\Omega}^p \approx 0$). That is, the final rotation estimate, which is accurate during both active and passive movements, is carried by different nodes (thus, likely different cell types; see Discussion) within the neural network.

When rotations change orientation relative to gravity, another internal state (tilt position $G$, not included in the simulations of *Figures 3* and *4*) and another sensor (otolith organs; $F = G$ since $A = 0$ in this simulation; *Figure 5F*, black) are engaged. During actively generated tilt movements, the rotation motor command ($\Omega^u$) is temporally integrated by the internal model (see *Eq. 3c* of Supplementary methods, 'Kalman filter algorithm developed'), generating an accurate prediction of head tilt $\hat{G}^p(t) = \int_0^t \Omega^u.dt$ (*Figure 5H*, left panel, green). This results in a correct prediction of the otolith signal $\hat{F}^p$ (*Figure 5I*, grey) and therefore, as in previous simulations of active movement, the sensory mismatch for both the canal and otolith signals (*Figure 5L*, magenta and gray, respectively) and feedback signals (not shown) are null; and the final estimates, driven exclusively by the prediction, are accurate; $\hat{G}(t) = \hat{G}^p(t)$ and $\hat{\Omega}(t) = \hat{\Omega}^p(t)$.

During passive tilt, the canal error, $\delta V$, is converted into Kalman feedback that updates $\hat{\Omega}$ (*Figure 5K*, blue) and $\hat{C}$ (not shown here; but see *Figure 5—figure supplement 1* for 2 s tilt simulations), as well as the two other state variables ($\hat{G}$ and $\hat{A}$). Specifically, the feedback from $\delta V$ to $\hat{G}$ ($G^k$) updates the predicted tilt $\hat{G}^p$ and is temporally integrated by the Kalman filter ($\hat{G}(t) = \int_0^t G^k$; see Supplementary methods, 'Passive Tilt'; *Figure 5K*, green). The feedback signal from $\delta V$ to $\hat{A}$ has a minimal impact, as illustrated in *Figure 5K*, red (see also Supplementary methods,' Kalman feedback gains' and *Table 2*).

Because $\delta V$ efficiently updates the tilt estimate $\hat{G}$, the otolith error $\delta F$ is close to zero during passive tilt (*Figure 5L*, gray; see Supplementary methods, 'Passive Tilt') and therefore all feedback signals originating from $\delta F$ (*Figure 5J*) play a minimal role (see Supplementary methods, 'Passive Tilt') during pure tilt (this is the case even for longer duration stimuli; *Figure 5—figure supplement 1*). This simulation highlights that, although tilt is sensed by the otoliths, passive tilt doesn't induce any sizeable otolith error. Thus, unlike neurons tuned to canal error, the model predicts that those cells tuned to otolith error will not modulate during either passive or actively-generated head tilt. Therefore, cells tuned to otolith error would respond primarily during translation, and not during tilt, thus they would be identified 'translation-selective'. Furthermore, the model predicts that those neurons tuned to passive tilt (e.g. Purkinje cells in the caudal cerebellar vermis; *Laurens et al., 2013b*) likely reflect a canal error that has been transformed into a tilt velocity error (*Figure 5L*, magenta). Thus, the model predicts that tilt-selective Purkinje cells should encode tilt velocity, and not tilt position, a prediction that remains to be tested experimentally (see Discussion).

## Otolith errors are interpreted as translation and tilt with distinct dynamics

Next, we simulate a brief translation (*Figure 6*). During active translation, we observe, as in previous simulations of active movements, that the predicted head motion matches the sensory (otolith in this case: $F = A$) signals ($\hat{A}^p = A$ and $\hat{F}^p = F$). Therefore, as in previous simulations of active motion, the sensory prediction error is zero (*Figure 6L*) and the final estimate is equal to, and driven by, the prediction ($\hat{A} = \hat{A}^p = A$; *Figure 6G*, red).

During passive translation, the predicted acceleration is null ($\hat{A}^p = 0$, *Figure 6H*, red), similar as during passive rotation in *Figures 3* and *4*. However, a sizeable tilt signal ($\hat{G}^p$ and $\hat{G}$, *Figure 6G,H*, green), develops over time. This (erroneous) tilt estimate can be explained as follows: soon after translation onset (vertical dashed lines in *Figure 6B–J*), $\hat{G}^p$ is close to zero. The corresponding predicted otolith signal is also close to zero ($\hat{F}^p = \hat{A}^p + \hat{G}^p = 0$), leading to an otolith error $\delta F \approx A$ (*Figure 6L*, right, gray). Through the Kalman feedback gain matrix, this otolith error, $\delta F$, is converted into: (1) an acceleration feedback $A^k$ (*Figure 6J*, red) with gain $k_{\delta F}^A = 0.995$ (the close to unity feedback gain indicates that otolith errors are interpreted as acceleration: $\hat{A} = \delta F \approx A$; note however that the otolith error $\delta F$ vanishes over time, as explained next); and (2) a tilt feedback $G^k$ (*Figure 6J*, green), with $k_{\delta F}^G = 0.5.\delta t$. This tilt feedback, although too weak to have any immediate effect, is integrated over time ($\hat{G}(t) = \int_0^t G^k$; see *Figure 5* and Supplementary methods, 'Somatogravic effect'), generating the rising tilt estimate $\hat{G}$ (*Figure 6G*, green) and $\hat{G}^p$ (*Figure 6H*, green).

The fact that the Kalman gain feedback from the otolith error to the $\hat{G}$ internal state generates the somatogravic effect is illustrated in *Figure 6—figure supplement 1*, where a longer acceleration

(20 s) is simulated. At the level of final estimates (perception), these simulations predict the occurrence of tilt illusions during sustained translation (somatogravic illusion; *Graybiel, 1952*; *Paige and Seidman, 1999*). Further simulations show how activation of the semicircular canals without a corresponding activation of the otoliths (e.g. during combination of tilt and translation; *Angelaki et al. (2004)*; *Yakusheva et al., 2007*) leads to an otolith error (*Figure 6—figure supplement 2*) and how signals from the otoliths (that sense indirectly whether or not the head rotates relative to gravity) can also influence the rotation estimate $\hat{\Omega}$ at low frequencies (*Figure 6—figure supplement 3*; this property has been extensively evaluated by *Laurens and Angelaki, 2011*). These simulations demonstrate that the Kalman filter model efficiently simulates all previous properties of both perception and neural responses during passive tilt and translation stimuli (see Discussion).

## Neck proprioceptors and encoding of trunk versus head velocity

The model analyzed so far has considered only vestibular sensors. Nevertheless, active head rotations often also activate neck proprioceptors, when there is an independent rotation of the head relative to the trunk. Indeed, a number of studies (*Kleine et al., 2004*; *Brooks and Cullen, 2009*; *2013*; *Brooks et al., 2015*) have identified neurons in the rostral fastigial nuclei that encode the rotation velocity of the trunk. These neurons receive convergent signals from the semicircular canals and neck muscle proprioception and, accordingly, are named 'bimodal neurons', to contrast with 'unimodal neurons', which encode passive head velocity. Because the bimodal neurons do not respond to active head and trunk movements (*Brooks and Cullen, 2013*; *Brooks et al., 2015*), they likely encode feedback signals related to trunk velocity. We developed a variant of the Kalman filter to model both unimodal and bimodal neuron types (*Figure 7*; see also Supplementary methods and *Figure 7—figure supplement 1–3*).

The model tracks the velocity of the trunk in space $\Omega_{TS}$ and the velocity of the head on the trunk $\Omega_{HT}$ as well as neck position ($N = \int \Omega_{HT}.dt$). Sensory inputs are provided by the canals (that sense the total head velocity, $\Omega = \Omega_{TS} + \Omega_{HT}$), and proprioceptive signals from the neck musculature ($P$), which are assumed to encode neck position (*Chan et al., 1987*).

In line with the simulations presented above, we find that, during active motion, the predicted sensory signals are accurate. Consequently, the Kalman feedback pathways are silent (*Figure 7—figure supplement 1–3*; active motion is not shown in *Figure 7*). In contrast, passive motion induces sensory errors and Kalman feedback signals. The velocity feedback signals (elaborated in *Figure 7—figure supplement 1–3*) have been re-plotted in *Figure 7*, where we illustrate head in space (blue), trunk in space (gray), and head on trunk (red) velocity (neck position feedback signals are only shown in *Figure 7—figure supplement 1–3*).

During passive whole head and trunk rotation, where the trunk rotates in space (*Figure 7A*, Real motion: $\Omega_{TS}>0$, grey) and the head moves together with the trunk (head on trunk velocity $\Omega_{HT} = 0$, red, head in space $\Omega>0$, blue), we find that the resulting feedback signals accurately encode these rotation components (*Figure 7A*, Velocity Feedback; see also *Figure 7—figure supplement 1*). During head on trunk rotation (*Figure 7B*, *Figure 7—figure supplement 2*), the Kalman feedback signals accurately encode the head on trunk (red) or in space (blue) rotation, and the absence of trunk in space rotation (gray). Finally, during trunk under head rotation that simulates a rotation of the trunk while the head remains fixed in space, resulting in a neck counter-rotation, the various motion components are accurately encoded by Kalman feedback (*Figure 7C*, *Figure 7—figure supplement 3*). We propose that unimodal and bimodal neurons reported in (*Brooks and Cullen, 2009*; *2013*) encode feedback signals about the velocity of the head in space ($\Omega^k$, *Figure 7*, blue) and of the trunk in space ($\Omega^k_{TS}$, *Figure 7*, gray), respectively. Furthermore, in line with experimental findings (*Brooks and Cullen, 2013*), these feedback pathways are silent during self-generated motion.

The Kalman filter makes further predictions that are entirely consistent with experimental results. First, it predicts that proprioceptive error signals during passive neck rotation encode velocity (*Figure 7—figure supplement 3L*; see Supplementary methods, 'Feedback signals during neck movement'). Thus, the Kalman filter explains the striking result that the proprioceptive responses of bimodal neurons encode trunk *velocity* (*Brooks and Cullen, 2009*; *2013*), even if neck proprioceptors encode neck position. Note that neck proprioceptors likely encode a mixture of neck position and velocity at high frequencies (*Chan et al., 1987*; *Mergner et al., 1991*); and additional simulations (not shown) based on this hypothesis yield similar results as those shown here. We used here a

model in which neck proprioceptors encode position for simplicity, and in order to demonstrate that Kalman feedback signals encode trunk velocity even when proprioceptive signals encode position.

Second, the model predicts another important property of bimodal neurons: their response gains to both vestibular (during sinusoidal motion of the head and trunk together) and proprioceptive (during sinusoidal motion of the trunk when the head is stationary) stimulation vary identically if a constant rotation of the head relative to the trunk is added, as an offset, to the sinusoidal motion (*Brooks and Cullen, 2009*). We propose that this offset head rotation extends or contracts individual neck muscles and affects the signal to noise ratio of neck proprioceptors. Indeed, simulations shown in *Figure 7—figure supplement 4* reproduce the effect of head rotation offset on bimodal neurons. In agreement with experimental findings, we also find that simulated unimodal neurons are not affected by these offsets (*Figure 7—figure supplement 4*).

Finally, the model also predicts the dynamics of trunk and head rotation perception during long-duration rotations (*Figure 7—figure supplement 5*), which has been established by behavioral studies (*Mergner et al., 1991*).

## Alternative models of interaction between active and passive motions

The theoretical framework of the Kalman filter asserts that the brain uses a single internal model to process copies of motor commands and sensory signals. But could alternative computational schemes, involving distinct internal models for motor and sensory signals, explain neuronal and behavioral responses during active and passive motions? Here, we consider three possibilities, illustrated in *Figure 1—figure supplement 1*. First, that the brain computes head motion based on motor commands only and suppresses vestibular sensory inflow entirely during active motion (*Figure 1—figure supplement 1A*). Second, that a 'motor' internal model and a 'sensory' internal model run in parallel, and that central neurons encode the difference between their outputs – which would represent a motion prediction error instead of a sensory prediction error, as proposed by the Kalman filter framework (*Figure 1—figure supplement 1B*). Third, that the brain computes sensory prediction errors based on sensory signals and the output of the 'motor' internal model, and then feeds these errors into the 'sensory' internal model (*Figure 1—figure supplement 1C*).

We first consider the possibility that the brain simply suppresses vestibular sensory inflow. Experimental evidence against this alternative comes from recordings performed when passive motion is applied concomitantly to an active movement (*Brooks and Cullen, 2013*; *2014*; *Carriot et al., 2013*). Indeed, neurons that respond during passive but not active motion have been found to encode the passive component of combined passive and active motions, as expected based on the Kalman framework. We present corresponding simulation results in *Figure 8*. We simulate a rotation movement (*Figure 8A*), where an active rotation ($\Omega^{\mathrm{u}}$, Gaussian velocity profile) is combined with a passive rotation ($\Omega^{\varepsilon}$, trapezoidal profile), a tilt movement (*Figure 8B*; using similar velocity inputs, $\Omega^{\mathrm{u}}$ and $\Omega^{\varepsilon}$, where the resulting active and passive tilt components are $\int \Omega^{\mathrm{u}} dt$ and $\int \Omega^{\varepsilon} dt$), and a translation movement (*Figure 8C*). We find that, in all simulations, the final motion estimate (*Figure 8D–F*; $\hat{\Omega}$, $\hat{G}$ and $\hat{A}$, respectively) matches the combined active and passive motions ($\Omega$, $G$ and $A$, respectively). In contrast, the Kalman feedback signals (*Figure 8G–I*) specifically encode the passive motion components. Specifically, the rotation feedback ($\Omega^{k}$, *Figure 8G*) is identical to the passive rotation $\Omega^{\varepsilon}$ (*Figure 8A*). As in *Figure 5*, the tilt feedback ($G^{k}$, *Figure 8H*) encodes tilt velocity, also equal to $\Omega^{\varepsilon}$ (*Figure 8A*). Finally, the linear acceleration feedback ($A^{k}$, *Figure 8I*) follows the passive acceleration component, although it decreases slightly with time because of the somatogravic effect. Thus, Kalman filter simulations confirm that neurons that encode sensory mismatch or Kalman feedback should selectively follow the passive component of combined passive and active motions.

What would happen if, instead of computing sensory prediction errors, the brain simply discarded vestibular sensory (or feedback) signals during active motion? We repeat the simulations of *Figure 8A–I* after removing the vestibular sensory input signals from the Kalman filter. We find that the net motion estimates encode only the active movement components (*Figure 8J–L*; $\hat{\Omega}$, $\hat{G}$ and $\hat{A}$) – thus, not accurately estimating the true movement. Furthermore, as a result of the sensory signals being discarded, all sensory errors and Kalman feedback signals are null. These simulations indicate that suppressing vestibular signals during active motion would prevent the brain from detecting passive motion occurring during active movement (see Discussion, 'Role of the vestibular system during

active motion: ecological, clinical and fundamental implications."), in contradiction with experimental results.

Next, we simulate (*Figure 9*) the alternative model of *Figure 1—figure supplement 1B*, where the motor commands are used to predict head motion (*Figure 9*, first row) while the sensory signals are used to compute a self-motion estimate (second row). According to this model, these two signals would be compared to compute a motion prediction error instead of a sensory prediction error (third row; presumably represented in the responses of central vestibular neurons). We first simulate short active and passive rotations (*Figure 9A,B*; same motion as in *Figure 3*). During active rotation (*Figure 9A*), both the motor prediction and the sensory self-motion estimate are close to the real motion and therefore the motor prediction is null (*Figure 9A*, third row). In contrast, the sensory estimate is not cancelled during passive rotation, leading to a non-zero motion prediction error (*Figure 9B*, third row). Thus, the motion prediction errors in *Figure 9A,B* resemble the sensory prediction errors predicted by the Kalman filter in *Figure 3* and may explain neuronal responses recorded during brief rotations.

However, this similarity breaks down when simulating a long-duration active or passive rotation (*Figure 9C,D*; same motion as in *Figure 4—figure supplement 1A,B*). The motor prediction of rotation velocity would remain constant during 1 min of active rotation (*Figure 9C*, first row), whereas the sensory estimate would decrease over time and exhibit an aftereffect (*Figure 9C*, second row). This would result in a substantial difference between the motor prediction and the sensory estimate (*Figure 9C*, third row) during active motion. This contrasts with Kalman filter simulations, where no sensory prediction errors occur during active motion.

A similar difference would also be seen during active translation (*Figure 9E*; same motion as in *Figure 6*). While the motion prediction (first row) would encode the active translation, the sensory estimate (second row) would be affected by the somatogravic effect (as in *Figure 6*), which causes the linear acceleration signal (red) to be replaced by a tilt illusion (green), also leading to motion prediction errors (third row). In contrast, the Kalman filter predicts that no sensory prediction error should occur during active translation.

These simulations indicate that processing motor and vestibular information independently would lead to prediction errors that would be avoided by the Kalman filter. Beyond theoretical arguments, this scheme may be rejected based on behavioral responses: Both rotation perception and the vestibulo-ocular reflex (VOR) decrease during sustained passive rotations, but persist indefinitely during active rotation (macaques: *Solomon and Cohen, 1992*); humans: *Guedry and Benson (1983)*; *Howard et al. (1998)*; *Jürgens et al., 1999*). In fact, this scheme cannot account for experimental findings, even if we consider different weighting for how the net self-motion signal is constructed from the independent motor and sensory estimates (*Figure 9*, bottom row). For example, if the sensory estimate is weighted 100%, rotation perception would decay during active motion (*Figure 9C*, bottom, dark blue), inconsistent with experimental results. If the motor prediction is weighted 100%, passive rotations would not be detected at all (*Figure 9B,D*, light blue). Finally, intermediate solutions (e.g. 50%/50%) would result in undershooting of both the steady state active (*Figure 9C*) and passive (*Figure 9B,D*) rotation perception estimates. Note also that, in all cases, the rotation aftereffect would be identical during active and passive motion (*Figure 9C,D*, bottom), in contradiction with experimental findings (*Solomon and Cohen, 1992*; *Guedry and Benson, 1983*; *Howard et al., 1998*).

Finally, the third alternative scheme (*Figure 1—figure supplement 1C*), where sensory prediction error is used to cancel the input of a sensory internal model is, in fact, a more complicated version of the Kalman filter. This is because an internal model that processes motor commands to predict sensory signals must necessarily include an internal model of the sensors. Thus, simulations of the model in *Figure 1—figure supplement 1C* would be identical to the Kalman filter, by merely re-organizing the sequence of operations and uselessly duplicating some of the elements, to ultimately produce the same results.

## Discussion

We have tested the hypothesis that the brain uses, during active motion, exactly the same sensory internal model computations already discovered using passive motion stimuli (*Mayne, 1974*; *Oman, 1982*; *Borah et al., 1988*; *Merfeld, 1995*; *Zupan et al., 2002*; *Laurens, 2006*; *Laurens and*

*Droulez, 2007*; *Laurens and Droulez, 2008*; *Laurens and Angelaki, 2011*; *Karmali and Merfeld, 2012*; *Lim et al., 2017*). Presented simulations confirm the hypothesis that the *same* internal model (consisting of forward internal models of the canals, otoliths and neck proprioceptors) can reproduce behavioral and neuronal responses to *both* active and passive motions. The formalism of the Kalman filter allows predictions of internal variables during both active and passive motions, with a strong focus on sensory error and feedback signals, which we hypothesize are realized in the response patterns of central vestibular neurons.

Perhaps most importantly, this work resolves an apparent paradox in neuronal responses between active and passive movements (*Angelaki and Cullen, 2008*), by placing them into a unified theoretical framework in which a single internal model tracks head motion based on motor commands and sensory feedback signals. Although particular cell types that encode sensory errors or feedback signals may not modulate during active movements because the corresponding sensory prediction error is negligible, the internal models of canal dynamics and otolith ambiguity operate continuously to generate the *correct sensory prediction* during *both* active and passive movements. Thus, the model presented here should eliminate the *misinterpretation that vestibular signals are ignored during self-generated motion, and that internal model computations during passive motion are unimportant for every day's life*. We hope that this realization should also highlight the relevance and importance of passive motion stimuli, as critical experimental paradigms that can efficiently interrogate the network and unravel computational principles of natural motor activities, which cannot easily be disentangled during active movements.

## Summary of the Kalman filter model

We have developed the first ever model that simulates self-motion estimates during both actively generated and passive head movements. This model, summarized schematically in *Figure 10*, transforms motor commands and Kalman filter feedback signals into internal estimates of head motion (rotation and translation) and predicted sensory signals. There are two important take-home messages: (1) Because of the physical properties of the two vestibular sense organs, the predicted motion generated from motor commands is not equal to predicted sensory signals (for example, the predicted rotation velocity is processed to account for canal dynamics in *Figure 4*). Instead, the predicted rotation, tilt and translation signals generated by efference copies of motor commands must be processed by the corresponding forward models of the sensors in order to generate accurate sensory predictions. This important insight about the nature of these internal model computations has not been appreciated by the qualitative schematic diagrams of previous studies. (2) In an environment devoid of externally generated passive motion, motor errors and sensory noise, the resulting sensory predictions would always match sensory afferent signals accurately. In a realistic environment, however, unexpected head motion occurs due to both motor errors and external perturbations (see 'Role of the vestibular system during active motion: ecological, clinical and fundamental implications'). Sensory vestibular signals are then used to correct internal motion estimates through the computation of sensory errors and their transformation into Kalman feedback signals. Given two sensory errors ($\delta V$ originating from the semicircular canals and $\delta F$ originating from the otoliths) and four internal state variables (rotation, internal canal dynamics, tilt and linear acceleration: $\hat{\Omega}, \hat{C}, \hat{G}, \hat{A}$), eight feedback signals must be constructed. However, in practice, two of these signals have negligible influence for all movements ($\delta V$ feedback to $\hat{A}$ and $\delta F$ feedback to $\hat{\Omega}$; see *Table 2* and Supplementary methods, 'Kalman Feedback Gains'), thus only six elements are summarized in *Figure 10*.

The non-negligible feedback signals originating from the canal error $\delta V$ are as follows (*Figure 10*, left):

- The feedback to the rotation estimate $\hat{\Omega}$ represents the traditional 'direct' vestibular pathway (*Raphan et al., 1979*; *Laurens and Angelaki, 2011*). It is responsible for rotation perception during high-frequency (unexpected) vestibular stimulation, and has a gain close to unity.
- The feedback to $\hat{C}$ feeds into the internal model of the canals, thus allowing compensation for canals dynamics. This pathway corresponds to the 'velocity storage' (*Raphan et al., 1979*; *Laurens and Angelaki, 2011*). Importantly, the contribution of this signal is significant for movements larger than ~1 s, particularly during high velocity rotations.

- The feedback to tilt ($\hat{G}$) converts canal errors into a tilt velocity ($dG/dt$) signal, which is subsequently integrated by the internal model of head tilt.

The non-negligible feedback signals originating from the otolith error $\delta F$ are as follows (*Figure 9*, right):

- The feedback to linear acceleration ($\hat{A}$) converts unexpected otolith activation into an acceleration signal and is responsible for acceleration perception during passive translations (as well as experimentally generated otolith errors; *Merfeld et al., 1999*; *Laurens et al., 2013a*).
- The $\delta F$ feedback to tilt ($\hat{G}$) implements the somatogravic effect that acts to bias the internal estimate of gravity toward the net otolith signal  so as to reduce the otolith error.
- The $\delta F$ feedback to $\hat{C}$ plays a similar role with the feedback to tilt $\hat{G}$, that is, to reduce the otolith error; but acts indirectly by biasing the internal estimate of rotation in a direction which, after integration, drives the internal model of tilt so that it matches otolith signal (this feedback was called 'velocity feedback' in *Laurens and Angelaki, 2011*). Behavioral studies (and model simulations) indicate that this phenomenon has low-frequency dynamics and results in the ability of otolith signals to estimate rotational velocity (*Angelaki and Hess, 1996*; *Hess and Angelaki, 1993*). Lesion studies have demonstrated that this feedback depends on an intact nodulus and ventral uvula, the vermal vestibulo-cerebellum (*Angelaki and Hess, 1995a*; *Angelaki and Hess, 1995b*).

The model in *Figure 10* is entirely compatible with previous models based on optimal passive self-motion computations (*Oman, 1982*; *Borah et al., 1988*; *Merfeld, 1995*; *Laurens, 2006*; *Laurens and Droulez, 2007*; *Laurens and Droulez, 2008*; *Laurens and Angelaki, 2011*; *Karmali and Merfeld, 2012*; *Lim et al., 2017*; *Zupan et al., 2002*). The present model is, however, distinct in two very important aspects: First, it takes into account active motor commands and integrates these commands with the vestibular sensory signals. Second, because it is formulated as a Kalman filter, it makes specific predictions about the feedback error signals, which constitute the most important nodes in understanding the neural computations underlying head motion sensation. Indeed, as will be summarized next, the properties of most cell types in the vestibular and cerebellar nuclei, as well as the vestibulo-cerebellum, appear to represent either sensory error or feedback signals.

## Vestibular and rostral fastigial neurons encode sensory error or feedback signals during rotation and translation

Multiple studies have reported that vestibular-only (erroneous term to describe 'non-eye-movement-sensitive') neurons in the VN encode selectively passive head rotation (*McCrea and Luan, 2003*; *Roy and Cullen, 2001*; *2004*; *Brooks and Cullen, 2014*) or passive translation (*Carriot et al., 2013*), but suppress this activity during active head movements. In addition, a group of rostral fastigial nuclei (unimodal rFN neurons; *Brooks and Cullen, 2013*; *Brooks et al., 2015*) also selectively encodes passive (but not active) rotations. These rotation-responding VN/rFN neurons likely encode either the semicircular canal error $\delta V$ itself or its Kalman feedback to the rotation estimate (blue in *Figure 10*, dashed and solid ovals 'VN, rFN', respectively). The translation-responding neurons likely encode either the otolith error $\delta F$ or its feedback to the linear acceleration estimate (*Figure 10*, solid and dashed red lines 'VN, trans PC'). Because error and feedback signals are proportional to each other in the experimental paradigms considered here, whether VN/rFN encode sensory errors or feedback signals cannot easily be distinguished using vestibular stimuli alone. Nevertheless, it is also important to emphasize that, while the large majority of VN and rFN neurons exhibit reduced responses during active head movements, this suppression is rarely complete (*McCrea et al., 1999*; *Roy and Cullen, 2001*; *Brooks and Cullen, 2013*; *Carriot et al., 2013*). Thus, neuronal responses likely encode mixtures of error/feedback and sensory motion signals (e.g. such as those conveyed by direct afferent inputs).

During large amplitude passive rotations (*Figure 4—figure supplement 3*), the rotation estimate persists longer than the vestibular signal (*Figure 4*, blue; a property called velocity storage). Because the internal estimate is equal to the canal error, this implies that VN neurons (that encode the canal error) should exhibit dynamics that are different from those of canal afferents, having incorporated velocity storage signals. This has indeed been demonstrated in VN neurons during optokinetic stimulation (*Figure 4—figure supplement 1*; *Waespe and Henn, 1977*; *Yakushin et al., 2017*) and

rotation about tilted axes (*Figure 6—figure supplement 3*; *Reisine and Raphan, 1992*; *Yakushin et al., 2017*).

## Thalamus-projecting VN neurons possibly encode final motion estimates

Based on the work summarized above, the final estimates of rotation (*Figure 4G*) and translation (*Figure 6G*), which are the desirable signals to drive both perception and spatial navigation, do not appear to be encoded by most VN/rFN cells. Thus, one may assume that they are reconstructed downstream, perhaps in thalamic (*Marlinski and McCrea, 2008*; *Meng et al., 2007*; *Meng and Angelaki, 2010*) or cortical areas. Interestingly, more than half (57%) of ventral thalamic neurons (*Marlinski and McCrea, 2008*) and an identical fraction (57%) of neurons of the VN cells projecting to the thalamus (*Marlinski and McCrea, 2009*) respond similarly during passive and actively-generated head rotations. The authors emphasized that VN neurons with attenuated responses during actively-generated movements constitute only a small fraction (14%) of those projecting to the thalamus. Thus, although abundant in the VN, these passive motion-selective neurons may carry sensory error/feedback signals to the cerebellum, spinal cord or even other VN neurons (e.g. those coding the final estimates; *Marlinski and McCrea, 2009*). Note that *Dale and Cullen, 2016*, reported contrasting results where a large majority of ventral thalamus neurons exhibit attenuated responses during active motion. Even if not present in the ventral posterior thalamus, this signal should exist in the spatial perception/spatial navigation pathways. Thus, future studies should search for the neural correlates of the final self-motion signal. VN neurons identified physiologically to project to the cervical spinal cord do not to modulate during active rotations, so they could encode either passive head rotation or active and passive trunk rotation (*McCrea et al., 1999*).

Furthermore, the dynamics of the thalamus-projecting VN neurons with similar responses to passive and active stimuli were not measured (*Marlinski and McCrea, 2009*). Recall that the model predicts that final estimates of rotation differ from canal afferent signals only in their response dynamics (*Figure 4*, compare panels F and G). It would make functional sense that these VN neurons projecting to the thalamus follow the final estimate dynamics (i.e., they are characterized by a prolonged time constant compared to canal afferents) – and future experiments should investigate this hypothesis.

## Rostral fastigial neurons encoding passive trunk rotations

Another class of rFN neurons (and possibly VN neurons projecting to the thalamus; *Marlinski and McCrea, 2009*, or those projecting to the spinal cord; *McCrea et al., 1999*) specifically encodes passive trunk velocity in space, independently of head velocity (bimodal neurons; *Brooks and Cullen, 2009*; *2013*; *Brooks et al., 2015*). These neurons likely encode Kalman feedback signals about trunk velocity (*Figure 7*, blue). Importantly, these neurons respond equivalently to passive whole trunk rotation when the trunk and the head rotate together (*Figure 7A*) and to passive trunk rotation when the head is space-fixed (*Figure 7C*). The first protocol activates the semicircular canals and induces a canal error $\delta V$, while the later activates neck proprioceptors and generates a proprioceptive error, $\delta P$. From a physiological point of view, this indicates that bimodal neurons respond to semicircular canals as well as neck proprioceptors (hence their name). Note that several other studies identified VN (*Anastasopoulos and Mergner, 1982*), rFN (*Kleine et al., 2004*) and anterior suprasylvian gyrus (*Mergner et al., 1985*) neurons that encode trunk velocity during passive motion, but didn't test their response to active motion.

The Kalman filter also predicts that neck proprioceptive signals that encode neck position should be transformed into error signals that encode neck velocity. In line with model predictions, bimodal neurons encode velocity signals that originate from neck proprioception during passive sinusoidal (1 Hz, *Brooks and Cullen, 2009*) and transient (Gaussian velocity profile, *Brooks and Cullen, 2013*) movements. Remarkably, although short-duration rotation of the trunk while the head is stationary in space leads to a veridical perception of trunk rotation, long duration trunk rotation leads to an attenuation of the perceived trunk rotation and a growing illusion of head rotation in the opposite direction (*Mergner et al., 1991*). These experimental findings are also predicted by the Kalman filter model (*Figure 7—figure supplement 5*).

## Purkinje cells in the vestibulo-cerebellum encode tilt and acceleration feedback

The simple spike modulation of two distinct types of Purkinje cells in the caudal cerebellar vermis (lobules IX-X, Uvula and Nodulus) encodes tilt (tilt-selective cells) and translation (translation-selective cells) during three-dimensional motion (*Yakusheva et al., 2007*, *2008*, *2013*; *Laurens et al., 2013a*; *Laurens et al., 2013b*). Therefore, it is possible that tilt- and translation selective cells encode tilt and acceleration feedback signals (*Figure 10*, green and red lines, respectively). If so, we hypothesize that their responses are suppressed during active motion (*Figures 5* and *6*). How Purkinje cells modulate during active motion is currently unknown. However, one study (*Lee et al., 2015*) performed when rats learned to balance on a swing indicates that Purkinje cell responses that encode trunk motion are reduced during predictable movements, consistent with the hypothesis that they encode sensory errors or Kalman feedback signals.

Model simulations have also revealed that passive tilt does not induce any significant otolith error (*Figure 5J*). In contrast, passive tilt elicits a significant canal error (*Figure 5K*). Thus, we hypothesize that the tilt signal present in the responses of Purkinje cells originates from the canal error $\delta V$ onto the tilt internal state variable. If it is indeed a canal, rather than an otolith, error, it should be proportional to tilt velocity instead of tilt position (or linear acceleration). Accordingly, we observed (*Laurens et al., 2013b*) that tilt-selective Purkinje cell responses were on average close to velocity (average phase lag of 36° during sinusoidal tilt at 0.5 Hz). However, since sinusoidal stimuli are not suited for establishing dynamics (*Laurens et al., 2017*), further experiments are needed to confirm that tilt-selective Purkinje cells indeed encode tilt velocity.

Model simulations have also revealed that passive translation, unlike passive tilt, should include an otolith error. This otolith error feeds also into the tilt internal variable (*Figure 9*, somatogravic feedback) and is responsible for the illusion of tilt during sustained passive linear acceleration (somatogravic effect; *Graybiel, 1952*). Therefore, as summarized in *Figure 10* (green lines), both canal and otolith errors should feedback onto the tilt internal variable. The canal error should drive modulation during tilt, whereas the otolith error should drive modulation during translation. In support of these predictions, we have demonstrated that tilt-selective Purkinje cells also modulate during translation, with a gain and phase consistent with the simulated otolith-driven feedback (*Laurens et al., 2013b*). Thus, both of these feedback error signals might be carried by caudal vermis Purkinje cells – and future experiments should address these predictions.

Note that semicircular canal errors must be spatially transformed in order to produce an appropriate tilt feedback. Indeed, converting a rotation into head tilt requires taking into account the angle between the rotation axis and earth-vertical. This transformation is represented by a bloc marked '3D' in *Figure 9* (see also (*eq. 9*) in Supplemenatry methods, 'Three-Dimensional Kalman filter'. Importantly, we have established (*Laurens et al., 2013b*) that tilt-selective Purkinje cells encode spatially transformed rotation signals, as predicted by theory. In fact, we have demonstrated that tilt-selective Purkinje cells do not simply modulate during vertical canal stimulation, but also carry the tilt signal during off-vertical axis yaw rotations (*Laurens et al., 2013b*).

In this respect, it is important to emphasize that truly tilt-selective neurons exclusively encode changes in orientation relative to gravity, rather than being generically activated by vertical canal inputs. Thus, it is critical that this distinction is experimentally made using three-dimensional motion (see *Laurens et al., 2013b*; *Laurens and Angelaki, 2015*). Whereas 3D rotations have indeed been used to identify tilt-selective Purkinje cells in the vermis (*Laurens et al., 2013b*; *Yakusheva et al., 2007*), this is not true for other studies. For example, *Siebold et al., 1997*, *Siebold et al., 1999*, *2001*), *Laurens and Angelaki, 2015* and *Zhou et al. (2006)* have reported tilt-modulated cells in the rFN and VN, respectively, but because these neurons were not tested in three dimensions, the signals carried by these neurons remain unclear.

## Further notes on tilt-selective Purkinje cells

As summarized above, the simple spike responses of tilt-selective Purkinje cells during passive motion have already revealed many details of the internal model computations. Thus, we have proposed that tilt- selective Purkinje cells encode the feedback signals about tilt, which includes scaled and processed (i.e. by a spatial transformation, green '3D' box in *Figure 10*) versions of both canal and otolith sensory errors (*Figure 10*, green oval, 'tilt PC?'). However, there could be alternative

implementations of the Kalman filter, where tilt-selective Purkinje cells may not encode only feedback signals, as proposed next:

We note that motor commands $\Omega^u$ must be also be spatially processed (black '3D' box in *Figure 10*) to contribute to the tilt prediction. One may question whether two distinct neuronal networks transform motor commands and canal errors independently (resulting in two '3D' boxes in *Figure 10*). An alternative (*Figure 10—figure supplement 1*) would be that the brain merges motor commands and canal error to produce a final rotation estimate prior to performing this transformation. From a mathematical point of view, this alternative would only require a re-arrangement of the Kalman filter equations, which would not alter any of the model's conclusions. However, tilt-selective Purkinje cells, which encode a spatially transformed signal, would then carry a mixture of predictive and feedback signals and would therefore respond identically to active and passive tilt velocity. Therefore, the brain may perform a spatial transformation of predictive and feedback rotation signals independently (*Figure 10*); or may merge them before transforming them (*Figure 10—figure supplement 1*). Recordings from tilt-selective Purkinje cells during active movements will distinguish between these alternatives.

## Summary of the neural implementation of sensory error and feedback signals

In summary, many of the response properties described by previous studies for vestibular nuclei and cerebellar neurons can be assigned a functional 'location' within the Kalman filter model. Interestingly, most of the central neurons fit well with the properties of sensory errors and/or feedback signals. That an extensive neural machinery has been devoted to feedback signals is not surprising, given their functional importance for self-motion estimation. For many of these signals, a distinction between sensory errors and feedback signals is not easily made. That is, rotation-selective VN and rFN neurons can encode either canal error (*Figure 10*, bottom, dashed blue oval) or rotation feedback (*Figure 10*, bottom, solid blue oval). Similarly, translation-selective VN, rFN and Purkinje cells can encode either otolith error (*Figure 10*, bottom, dashed red oval) or translation feedback (*Figure 10*, bottom, solid red oval). The only feedback that is easily distinguished based on currently available data is the tilt feedback (*Figure 10*, green lines).

Although the blue, green and red feedback components of *Figure 10* can be assigned to specific cell groups, this is not the case with the cyan feedback components. First, note that, like the tilt variable, the canal internal model variable, receives non-negligible feedback contributions from both the canal and otolith sensory errors (*Figure 10*, cyan lines). The canal feedback error changes the time constant of the rotation estimate (*Figure 4* and *Figure 4—figure supplements 1* and *3*), whereas the otolith feedback error may suppress (post-rotatory tilt) or create (horizontal axis rotation) a rotation estimate (*Figure 6—figure supplement 3*). The neuronal implementations of the internal model of the canals ($\hat{C}$), and of its associated feedback pathways, are currently unknown. However, lesion studies clearly indicate that the caudal cerebellar vermis, lobules X and IX may influence the canal internal model state variable (*Angelaki and Hess, 1995a*; *Angelaki and Hess, 1995b*; *Wearne et al., 1998*). In fact, it is possible that the simple-spike output of the translation-selective Purkinje cells also carries the otolith sensory error feedback to the canal internal model state variable (*Figure 10*, bottom, cyan arrow passing though the dashed red ellipse). Similarly, the canal error feedback to the canal internal model state variable (*Figure 10*, bottom, cyan arrow originating from the dashed blue ellipse) can originate from VN or rFN cells that selectively encode passive, not active, head rotation (*Figure 4J*, note that the $C^k$ feedback is but a scaled-down version of the $\Omega^k$ feedback).

Thus, although the feedback error signals to the canal internal model variable can be linked to known neural correlates, cells coding for the state variable $\hat{C}$ exclusively have not been identified. It is possible that the hidden variable $\hat{C}$ may be coded in a distributed fashion. After all, as already stated above, VN and rFN neurons have also been shown to carry mixed signals - they can respond to both rotation and translation, as well as they may carry both feedback/error and actual sensory signals. Thus, it is important to emphasize that these Kalman variables and error signals may be represented in a multiplexed way, where single neurons manifest mixed selectivity to more than just one internal state and/or feedback signals. This appears to be an organizational principle both in central vestibular areas (*Laurens et al., 2017*) and throughout the brain (*Rigotti et al., 2013*;

*Fusi et al., 2016*). It has been proposed that mixed selectivity has an important computational advantage: high-dimensional representations with mixed selectivity allow a simple linear readout to generate a diverse array of potential responses (*Fusi et al., 2016*). In contrast, representations based on highly specialized neurons are low dimensional and may preclude a linear readout from generating several responses that depend on multiple task-relevant variables.

## Recalibration of motor internal model computations during proprioceptive mismatch

In this treatment, we have considered primarily the importance of the internal models of the sensors to emphasize its necessity for both self-generated motor commands and unpredicted, external perturbations. It is important to point out that self-generated movements involve internal model computations that have been studied extensively in the field of motor control and motor adaptation (*Wolpert et al., 1995*; *Körding and Wolpert, 2004*; *Todorov, 2004*; *Chen-Harris et al., 2008*; *Berniker et al., 2010*; *Berniker and Kording, 2011*; *Franklin and Wolpert, 2011*; *Saglam et al., 2011*; *2014*). While the question of motor adaptation are not addressed directly in the present study, experiments in which resistive or assistive torques are applied to the head (*Brooks et al., 2015*) or in which active movements are entirely blocked (*Roy and Cullen, 2004*; *Carriot et al., 2013*) reveal how central vestibular pathways respond in situations that cause motor adaptation. Under these conditions, central neurons have been shown to encode net head motion (i.e. active and passive indiscriminately) with a similar gain as during passive motion (*Figure 7—figure supplements 6* and *7*). This may be interpreted and modeled by assuming that central vestibular pathways cease to integrate copies of motor commands (*Figure 7—figure supplement 6*) whenever active head motion is perturbed, until the internal model of the motor plant recalibrates to anticipate this perturbation (*Brooks et al., 2015*). Further analysis of these experimental results (*Figure 7—figure supplement 7*) indicate that they are fundamentally non-linear and cannot be reproduced by the Kalman filter (which is limited to linear operations) and therefore requires the addition of an external gating mechanism (black pathway in *Figure 1D*).

Notably, this nonlinearity is triggered with proprioceptive mismatch, that is, when there is a discrepancy between the intended head position and proprioceptive feedback. Note that perturbing head motion also induces a vestibular mismatch since it causes the head velocity to differ from the motor plan. However, central vestibular neurons still encode specifically passive head movement during vestibular mismatch, as can be shown by superimposing passive whole body rotations to active movements (*Brooks and Cullen, 2013*; *2014*; *Carriot et al., 2013*) and illustrated in the model predictions of *Figure 8*. Remarkably, the elementary and fundamental difference between these different types of computations has never before been presented in a single theoretical framework.

Proprioceptive mismatch is likely a specific indication that the internal model of the motor plant (necessary for accurate motor control; *Figure 1D*, red) needs to be recalibrated. Applying resistive head torques (*Brooks et al., 2015*) or increasing head inertia (*Saglam et al., 2011*; *2014*) does indeed induce motor adaptation which is not modeled in the present study (but see *Berniker and Kording, 2008* ). Interestingly, the studies by *Saglam et al. (2011)*, *2014* indicate that healthy subjects use a re-calibrated model of the motor plant to restore optimal motor performance, but that vestibular deficient patients fail to do so, indicating that vestibular error signals participate in motor adaptation (*Figure 1D*, broken blue arrow).

## Relation to previous dynamical models

The internal model framework has been widely used to simulate optimal motor control strategies (*Todorov, 2004*; *Chen-Harris et al., 2008*; *Saglam et al., 2011*; *2014*) and to create Kalman filter models of reaching movements (*Berniker and Kording, 2008*) and postural control (*van der Kooij et al., 2001*). The present model, however, is to our knowledge the first to apply these principles to optimal head movement perception during active and passive motion. As such, it makes explicit links between sensory dynamics (i.e. the canals), ambiguities (i.e. the otoliths), priors and motor efference copies. Perhaps most importantly, the focus of this study has been to explain neuronal response properties. By simulating and explaining neuronal responses during active and passive self-motion in the light of a quantitative model, this study advances our understanding of how theoretical principles

about optimal combinations of motor signals, multiple sensory modalities with distinct dynamic properties and ambiguities and Bayesian priors map onto brainstem and cerebellar circuits.

To simplify the main framework and associated predictions, as well as the in-depth mathematical analyses of the model's dynamics (Supplementary methods), we have presented a linearized one-dimensional model. This model was used to simulate either rotations around an earth-vertical axis or combinations of translation and rotations around an earth-horizontal axis. A more natural and general way to simulate self-motion information processing is to design a three-dimensional Kalman filter model. Such models have been used in previous studies, either by programming Kalman filters explicitly (*Borah et al., 1988*; *Lim et al., 2017*), or by building models based on the Kalman filter framework (*Glasauer, 1992*; *Merfeld, 1995*; *Glasauer and Merfeld, 1997*; *Bos et al., 2001*; *Zupan et al., 2002*). We show in Supplementary methods, 'Three-dimensional Kalman filter', how to generalize the model to three dimensions.

The passive motion components of the model presented here are to a large extent identical to the Particle filter Bayesian model in (*Laurens, 2006*; *Laurens and Droulez, 2007*,*Laurens and Droulez, 2008*; *Laurens and Angelaki, 2011*), which we have re-implemented as a Kalman filter, and into which we incorporated motor commands. One fundamental aspect of previous Bayesian models (*Laurens, 2006*; *Laurens and Droulez, 2007*,*Laurens and Droulez, 2008*) is the explicit use of two Bayesian priors that prevent sensory noise from accumulating over time. These priors encode the natural statistics of externally generated motion or motion resulting from motor errors and unexpected perturbations. Because, on average, rotation velocities and linear accelerations are close to zero, these Bayesian priors are responsible for the decrease of rotation estimates during sustained rotation (*Figure 4—figure supplement 2*) and for the somatogravic effect (*Figure 6—figure supplement 2*) (see *Laurens and Angelaki, 2011*) for further explanations). The influence of the priors is higher when the statistical distributions of externally generated rotation ($\Omega^e$) and acceleration ($A^e$) are narrower (*Figure 10—figure supplement 2*), that is when their standard deviation is smaller. Stronger priors reduce the gain and time constant of rotation and acceleration estimates (*Figure 10—figure supplement 2B,D*). Importantly, the Kalman filter model predicts that the priors affect only the passive, but not the active, self-motion final estimates. Indeed, the rotation and acceleration estimates last indefinitely during simulated active motion (*Figure 4—figure supplement 2*, *Figure 6—figure supplement 2*, *Figure 10—figure supplement 2*). In this respect, the Kalman filter may explain why the time constant of the vestibulo-ocular reflex is reduced in figure ice skaters (*Tanguy et al., 2008*; *Alpini et al., 2009*): The range of head velocities experienced in these activities is wider than normal. In previous Bayesian models, we found that widening the rotation prior should increase the time constant of vestibular responses, apparently in contradiction with these experimental results. However, these models did not consider the difference between active and passive stimuli. The formalism of the Kalman filter reveals that Bayesian priors should reflect the distribution of passive motion or motor errors. In athletes that are highly trained to perform stereotypic movements, this distribution likely narrows, resulting in stronger priors and reduced vestibular responses.

## Further behavioral evidence of optimal combination of vestibular signals and efference copies

One of the predictions of the Kalman filter is that motion illusions, such as the disappearance of rotation perception during long-duration rotation and the ensuing post-rotatory response (*Figure 4—figure supplement 1B*) should not occur during active motion (*Figure 4—figure supplement 1A*). This has indeed been observed in monkeys (*Solomon and Cohen, 1992*) and humans, where steady-state per-rotatory responses plateau at 10°/s and post-rotatory responses are decreased by a similar amount (*Guedry and Benson, 1983*; *Howard et al., 1998*); see also *Brandt et al., 1977a*). The fact that post-rotatory responses are reduced following active, as compared to passive, rotations is of particular interest, because it demonstrates that motor commands influence rotation perception even after the movement has stopped. As shown in *Figure 4*, The Kalman filter reproduces this effect by feeding motor commands though an internal model of the canals. As shown in *Figure 4—figure supplement 1*, this process is equivalent to the concept of 'velocity storage' (*Raphan et al., 1979*; see *Laurens, 2006*, *Laurens and Droulez, 2008*, *MacNeilage et al. (2008)*, *Laurens and Angelaki (2011)* for a Bayesian interpretation of the concept of velocity storage). Therefore, the functional significance of this network, including velocity storage, is found during natural active head

movements (see also *Figure 4—figure supplement 3*), rather than during passive low-frequency rotations with which it has been traditionally associated with in the past (but see *Laurens and Angelaki, 2011*).

A recent study (*MacNeilage and Glasauer, 2017*) evaluated how motor noise varies across locomotor activities and within gait cycles when walking. They found that motor noise peaks shortly before heel strike and after toe off; and is minimal during swing periods. They interpreted experimental findings using principles of sensory fusion, an approach that uses the same principles of optimal cue combination as the Kalman filter but doesn't include dynamics. Interestingly, this analysis showed that vestibular cues should have a maximal effect when motor noise peaks, in support with experimental observations (*Brandt et al., 1999*; *Jahn et al., 2000*).

To avoid further complications to the solution to the Kalman filter gains, the presented model does not consider how the brain generates motor commands in response to vestibular stimulation, e.g. to stabilize the head in response to passive motion or to use vestibular signals to correct motor commands. This would require an additional feedback pathway - the reliance of motor command generation on sensory estimates (*Figure 1D*, blue broken arrow). For example, a passive head movement could result in a stabilizing active motor command. Or an active head movement could be less than desired because of noise, requiring an adjustment of the motor command to compensate. These feedback pathways have been included in previous Kalman filter models (e.g. *van der Kooij et al., 2001*), a study that focused specifically on postural control and reproduced human postural sway under a variety of conditions. Thus, the Kalman filter framework may be extended to model neuronal computations underlying postural control as well as the vestibulo-collic reflex.

## Role of the vestibular system during active motion: ecological, clinical and fundamental implications

Neuronal recordings (*Brooks and Cullen, 2013*; *2014*; *Carriot et al., 2013*) and the present modeling unambiguously demonstrate that central neurons respond to unexpected motion during active movement (a result that we reproduced in *Figure 8G–I*). Beyond experimental manipulations, a number of processes may cause unpredictable motion to occur in natural environments. When walking on tree branches, boulders or soft grounds, the support surface may move under the feet, leading to unexpected trunk motion. A more dramatic example of unexpected trunk motion, that requires immediate correction, occurs when slipping or tripping. Complex locomotor activities involve a variety of correction mechanism among which spinal mechanisms and vestibular feedback play preeminent roles (*Keshner et al., 1987*; *Black et al., 1988*; *Horstmann and Dietz, 1988*).

The contribution of the vestibular system for stabilizing posture is readily demonstrated by considering the impact of chronic bilateral vestibular deficits. While most patients retain an ability to walk on firm ground and even perform some sports (*Crawford, 1964*; *Herdman, 1996*), vestibular deficit leads to an increased incidence of falls (*Herdman et al., 2000*), difficulties in walking on uneven terrains and deficits in postural responses to perturbations (*Keshner et al., 1987*; *Black et al., 1988*; *Riley, 2010*). This confirms that vestibular signals are important during active motion, especially in challenging environments. In this respect, the Kalman filter framework appears particularly well suited for understanding the effect of vestibular lesions.

As mentioned earlier, vestibular sensory errors also occur when the internal model of the motor apparatus is incorrect (*Brooks et al., 2015*) and these errors can lead to recalibration of internal models. This suggests that vestibular error signals during self-generated motion may play two fundamental roles: (1) updating self-motion estimates and driving postural or motor corrections, and (2) providing teaching signals to internal models of motor control (*Wolpert et al., 1995*) and therefore facilitating motor learning. This later point is supported by the finding that patients with vestibular deficits fail to recalibrate their motor strategies to account for changes in head inertia (*Sağlam et al., 2014*).

But perhaps most importantly, the model presented here should eliminate the misinterpretation that vestibular signals are ignored during self-generated motion – and that passive motion stimuli are old-fashioned and should no longer be used in experiments. Regarding the former conclusion, the presented simulations highlight the role of the internal models of canal dynamics and otolith ambiguity, which operate continuously to generate the *correct sensory prediction* during *both* active and passive movements. Without these internal models, the brain would be unable to correctly predict sensory canal and otolith signals and everyday active movements would lead to sensory

mismatch (e.g. for rotations, see *Figure 4—figure supplements 2* and *3*). Thus, even though particular nodes (neurons) in the circuit show attenuated or no modulation during active head rotations, vestibular processing remains the same - the internal model is both engaged and critically important for accurate self-motion estimation, even during actively-generated head movements. Regarding the latter conclusion, it is important to emphasize that passive motion stimuli have been, and continue to be, extremely valuable in revealing salient computations that would have been amiss if the brain's intricate wisdom was interrogated only with self-generated movements.

Furthermore, a quantitative understanding of how efference copies and vestibular signals interact for accurate self-motion sensation is primordial for our understanding of many other brain functions, including balance and locomotor control. As stated in *Berniker and Kording (2011)*: 'A crucial first step for motor control is therefore to integrate sensory information reliably and accurately', and practically any locomotor activity beyond reaching movements in seated subjects will affect posture and therefore recruit the vestibular sensory modality. It is thus important for both motor control and spatial navigation functions (for which intact vestibular cues appear to be critical; *Taube, 2007*) to correct the misconception of incorrectly interpreting that vestibular signals are cancelled and thus are not useful during actively generated movements. By providing a state-of-the-art model of self-motion processing during active and passive motion, we are bridging several noticeable gaps between the vestibular and motor control/navigation fields.

## Conclusion

'A good model has a delightful way of building connections between phenomena that never would have occurred to one' (*Robinson, 1977*). Four decades later, this beautifully applies here, where the mere act of considering how the brain should process self-generated motion signals in terms of mathematical equations (instead of schematic diagrams) immediately revealed a striking similarity with models of passive motion processing and, by motivating this work, opened an avenue to resolve a standing paradox in the field.

The internal model framework and the series of quantitative models it has spawned have explained and simulated behavioral and neuronal responses to self-motion using a long list of passive motion paradigms, and with a spectacular degree of accuracy (*Mayne, 1974*; *Oman, 1982*; *Borah et al., 1988*; *Glasauer, 1992*; *Merfeld, 1995*; *Glasauer and Merfeld, 1997*; *Bos et al., 2001*; *Zupan et al., 2002*; *Laurens, 2006*; *Laurens and Droulez, 2007*,*Laurens and Droulez, 2008*; *Laurens and Angelaki, 2011*; *Karmali and Merfeld, 2012*; *Lim et al., 2017*). Internal models also represent the predominant theoretical framework for studying motor control (*Wolpert et al., 1995*; *Körding and Wolpert, 2004*; *Todorov, 2004*; *Chen-Harris et al., 2008*; *Berniker et al., 2010*; *Berniker and Kording, 2011*; *Franklin and Wolpert, 2011*; *Saglam et al., 2011*; *2014*). The vestibular system shares many common questions with the motor control field, such as that of 3D coordinate transformations and dynamic Bayesian inference, but, being considerably simpler, can be modeled and studied using relatively few variables. As a result, head movements represent a valuable model system for investigating the neuronal implementation of computational principles that underlie motor control. The present study thus offers the theoretical framework which will likely assist in understanding neuronal computations that are essential to active self-motion perception, spatial navigation, balance and motor activity in everyday life.

## Materials and methods

### Structure of a Kalman filter

In a Kalman filter (*Kalman, 1960*), state variables $X$ are driven by their own dynamics (matrix $D$), motor commands $X^u$ and unpredictable perturbations resulting from motor noise and external influence $X^\varepsilon$ through the equation (*Figure 1—figure supplement 2A*):

$$X(t) = D.X(t-1) + M.X^u(t) + E.X^\varepsilon$$

where matrices $M$ and $E$ reflect the response to motor inputs and perturbations, respectively.

A set of sensors, grouped in a variable $S$, measure state variables transformed by a matrix $T$, and are modeled as:

$$S(t) = T.X(t) + S^\eta(t)$$

where $S^\eta$ is Gaussian sensory noise (**Figure 1—figure supplement 2A**, right). The model assumes that the brain has an exact knowledge of the forward model, that is, of $D$, $M$, $E$ and $T$ as well as the variances of $X^\varepsilon$ and $S^\eta$. Furthermore, the brain knows the values of the motor inputs $X^u$ and sensory signals $S$, but doesn't have access to the actual values of $X^\varepsilon$ and $S^\eta$.

At each time $t$, the Kalman filter computes a preliminary estimate (also called a prediction) $\hat{X}^p(t) = D.\hat{X}(t-1) + M. X^u(t)$ and a corresponding predicted sensory signal $\hat{S}^p(t) = T.\hat{X}^p(t)$ (**Figure 1—figure supplement 2B**). In general, the resulting state estimate $\hat{X}^p(t)$ and the predicted sensory prediction $\hat{S}^p(t)$ may differ from the real values $X(t)$ and $S(t)$ because: (1) $X^\varepsilon(t) \neq 0$, but the brain cannot predict the perturbation $X^\varepsilon(t)$, (2) the brain does not know the value of the sensory noise $S^\eta(t)$ and (3) the previous estimate $\hat{X}(t-1)$ used to compute $\hat{X}^p(t)$ could be incorrect. These errors are reduced using sensory information, as follows (**Figure 1—figure supplement 2B**). First, this prediction $\hat{S}^p(t)$ and the sensory input $S(t)$ are compared to compute a sensory error $\delta S(t)$. Second, sensory errors are then transformed into a feedback $X^k(t) = K.\delta S(t)$ where $K$ is a matrix of feedback gains. Thus, an improved estimate at time $t$ is $\hat{X}(t) = \hat{X}^p(t) + K.\delta S(t)$. The value of the feedback gain matrix $K$ determines how sensory errors (and therefore sensory signals) are used to compute the final estimate $\hat{X}(t)$ and is computed based on $D$, $E$, $T$ and on the variances of $X^\varepsilon$ and $S^\eta$ (see Supplementary methods, 'Kalman filter algorithm').

In the case of the self-motion model, the motor commands $\Omega^u$ and $A^u$ are inputs to the Kalman filter (**Figure 2**). Note that, while the motor system may actually control other variables (such as forces or accelerations), we consider that these variables are converted into $\Omega^u$ and $A^u$. We demonstrate in Supplementary methods,' Model of motor commands' that altering these assumptions does not alter our conclusions. In addition to motor commands, a variety of unpredictable factors such as motor noise and external (passive) motion also affect $\Omega$ and $A$ (**MacNeilage and Glasauer, 2017**). The total rotation and acceleration components resulting from these factors are modeled as variables $\Omega^\varepsilon$ and $A^\varepsilon$. Similar to (**Laurens, 2006**; **Laurens and Droulez, 2007**, **Laurens and Droulez, 2008**) we modeled the statistical distribution of these variables as Gaussians, with standard deviations $\sigma_\Omega$ and $\sigma_A$.

Excluding vision and proprioception, the brain senses head motion though the semicircular canals (that generate a signal $V$) and the otoliths organs (that generate a signal $F$). Thus, in initial simulations (**Figures 3–6**), the variable $S$ encompasses $V$ and $F$ (neck proprioceptors are added in **Figure 7**).

The semicircular canals are rotation sensors that, due to their mechanical characteristic, exhibit high-pass filter properties. These dynamics may be neglected for rapid movements of small amplitude (such as **Figure 3**) but can have significant impact during natural movements (**Figure 4—figure supplement 3**). They are modeled using a hidden state variable $C$. The canals are also subject to sensory noise $V^\eta$. Taken both the noise and the dynamics into account, the canals signal is modeled as $V = \Omega - C + V^\eta$.

The otolith organs are acceleration sensors. They exhibit negligible temporal dynamics in the range of motion considered here, but are fundamentally ambiguous: they sense gravitational as well as linear acceleration – a fundamental ambiguity resulting from Einstein's equivalence principle (**Einstein, 1907**). Gravitational acceleration along the inter-aural axis depends on head roll position, modeled here as $G = \int \Omega.dt$. The otoliths encode the sum of $A$ and $G$ and is also affected by sensory noise $F^\eta$, such that the net otolith signal is $F = A + G + F^\eta$.

How sensory errors are used to correct motion estimates depends on the Kalman gain matrix, which is computed by the Kalman algorithm such that the Kalman filter as a whole performs optimal estimation. In theory, the Kalman filter includes a total of 8 feedback signals, corresponding to the combination of two sensory errors (canal and otolith errors) and four internal states (see Supplementary methods,' Kalman feedback gains').

It is important to emphasize that the Kalman filter model is closely related to previous models of vestibular information processing. Indeed, simulations of long-duration rotation and visuo-vestibular interactions (**Figure 4—figure supplement 2**), as well as mathematical analysis (**Laurens, 2006**), demonstrate that $\hat{C}$ is equivalent to the 'velocity storage' (**Raphan et al., 1979**; **Laurens and Angelaki, 2011**). These low-frequency dynamics, as well as visuo-vestibular interactions, were previously

simulated and interpreted in the light of optimal estimation theory; and accordingly are reproduced by the Kalman filter model.

The model presented here is to a large extent identical to the Particle filter Bayesian model in (*Laurens, 2006*; *Laurens and Droulez, 2007*, *Laurens and Droulez, 2008*; *Laurens and Angelaki, 2011*). It should be emphasized that: (1) transforming the model into a Kalman filter didn't alter the assumptions upon which the Particle filter was build; (2) introducing motor commands into the Kalman filter was a textbook process that did not require any additional assumptions or parameters; and (3) we used exactly the same parameter values as in *Laurens, 2006* and *Laurens and Droulez, 2008* (with the exception of $\sigma_F$ whose impact, however, is negligible, and of the model of head on trunk rotation that required additional parameters; see next section).

## Simulation parameters

The parameters of the Kalman filter model are directly adapted from previous studies (*Laurens, 2006*; *Laurens and Droulez, 2008*). Tilt angles are expressed in radians, rotation velocities in rad/s, and accelerations in $g$ (1 $g$ = 9.81 m/s$^2$). Note that a small linear acceleration $A$ in a direction perpendicular to gravity will rotate the gravito-inertial force vector around the head by an angle $\alpha = sin^{-1}(A) \approx A$. For this reason, tilt and small amplitude linear accelerations are expressed, in one dimension, in equivalent units that may be added or subtracted. The standard deviations of the unpredictable rotations ($\Omega^\varepsilon$) and accelerations ($A^\varepsilon$) are set to the standard deviations of the Bayesian a priori in *Laurens, 2006* and *Laurens and Droulez, 2008*, that is, $\sigma_\Omega = $ 0.7 rad/s ($\Omega^\varepsilon$) and $\sigma_A$ =0.3 $g$ ($A^\varepsilon$). The standard deviation of the noise affecting the canals ($V^\eta$) was set to $\sigma_V$ =0.175 rad/s (as in *Laurens, 2006* and *Laurens and Droulez, 2008*; see *Figure 10—figure supplement 2* for simulations with different parameters). The standard deviation of the otolith noise ($F^\eta$) was set to $\sigma_F$ =0.002 $g$ (2 cm/s$^2$). We verified that values ranging from 0 to 0.01 $g$ had no effect on simulation results. The time constant of the canals was set to $\tau_c$=4s. Simulations used a time step of $\delta t$ = 0.01 s. We verified that changing the value of the time step without altering other parameters had no effect on the results.

We ran simulations using a variant of the model that included visual information encoding rotation velocity. The visual velocity signals were affected by sensory noise with a standard deviation $\sigma_{Vis}$ = 0.12 rad/s, as in *Laurens and Droulez, 2008*.

Another variant modeled trunk in space velocity ($\Omega_{TS}$) and head on trunk velocity ($\Omega_{HT}$) independently. The standard deviations of unpredictable rotations were set to $\sigma_{TS}$ = 0.7 rad/s (identical to $\sigma_\Omega$) and $\sigma_{HT}$ = 3.5 rad/s. The standard deviation of sensory noise affecting neck afferents was set manually to $\sigma_P$ = 0.0017 rad. We found that increasing the neck afferent noise reduces the gain of head on trunk and trunk in space velocity estimate (*Figure 7C*) (e.g. by 60% for a tenfold increase in afferent noise). Reducing the value of this noise has little effect on the simulations.

For simplicity, all simulations were run without adding the sensory noise $V^\eta$ and $F^\eta$. These noise-free simulations are representative of the results that would be obtained by averaging several simulation runs performed with sensory noise (e.g. as in *Laurens and Droulez, 2007*). We chose to present noise-free results here in order to facilitate the comparison between simulations of active and passive motions.

A Matlab implementation of the Kalman model is available at: https://github.com/JeanLaurens/Laurens_Angelaki_Kalman_2017 (*Laurens, 2017*; copy archived at https://github.com/elifesciences-publications/Laurens_Angelaki_Kalman_2017).

# Additional information

### Funding

| Funder | Grant reference number | Author |
|---|---|---|
| National Institute on Deafness and Other Communication Disorders | NIH R01DC004260 | Jean Laurens Dora E Angelaki |

The funders had no role in study design, data collection and interpretation, or the decision to submit the work for publication.

## Author contributions
Jean Laurens, Conceptualization, Investigation, Writing—original draft, Writing—review and editing; Dora E Angelaki, Conceptualization, Supervision, Funding acquisition, Writing—original draft, Writing—review and editing

## Author ORCIDs
Jean Laurens, http://orcid.org/0000-0002-9101-2802
Dora E Angelaki, http://orcid.org/0000-0002-9650-8962

## Decision letter and Author response
Decision letter https://doi.org/10.7554/eLife.28074.036
Author response https://doi.org/10.7554/eLife.28074.037

## Additional files

### Supplementary files
• Transparent reporting form
DOI: https://doi.org/10.7554/eLife.28074.034

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

# Appendix 1

DOI: https://doi.org/10.7554/eLife.28074.035

## Supplementary Methods

Here we describe the Kalman model in more detail. We present the model of head motion and vestibular information processing, first as a set of linear equations ('Model of head motion and vestibular sensors'), and then in matrix form ('Model of head motion in matrix form'). Next we present the Kalman filter algorithm, in the form of matrix computations ('Kalman filter algorithm') and then as a series of equations ('Kalman filter algorithm developed').

Next, we derive a series of properties of the internal model computations ('Velocity Storage during EVAR'; 'Passive Tilt',' Kalman feedback gains',' Time constant of the somatogravic effect',' Model of motor commands').

We then present some variations of the Kalman model ('Visual rotation signals',' Model of head and neck rotations', 'Feedback signals during neck movement', 'Three-dimensional Kalman filter').

## Model of head motion and vestibular sensors

The model of head motion in (*Figure 2*) can be described by the following equations (see *Table 1* for a list of mathematical variables):

$$\begin{cases} \Omega(t) &= \quad\quad 0 \quad\quad + \quad \Omega^u(t) \; + \; \Omega^\varepsilon(t) \quad (eq.\,1) \\ C(t) &= k_1.C(t-\delta t) + k_2.\Omega(t) \; + \quad 0 \quad + \quad 0 \quad (eq.\,2) \\ G(t) &= \quad G(t-\delta t) + s.\delta t.\Omega(t) \; + \quad 0 \quad + \quad 0 \quad (eq.\,3) \\ A(t) &= \quad\quad 0 \quad\quad + \quad A^u(t) \; + \; A^\varepsilon(t) \quad (eq.\,4) \end{cases}$$

Here, *eq.* 1 states that head velocity $\Omega(t)$ is the sum of self-generated rotation $\Omega^u(t)$ and of unexpected rotations $\Omega^\varepsilon(t)$ resulting from motor errors and passive motion. In the absence of motor commands, $\Omega(t)$ is expected to be zero on average, independently from all previous events.

*Eq.* 2 describes the first-order low-pass dynamics of the canals:

$$C(t) = C(t-\delta t) + \frac{\delta t}{\tau_c}.(\Omega(t) - C(t))$$

which yields:

$$C(t) = \frac{\tau_c}{\tau_c + \delta t}.C(t-\delta t) + \frac{\delta t}{\tau_c + \delta t}.\Omega(t) \quad (eq.\,2)$$

with $k_1 = \frac{\tau_c}{\tau_c + \delta t}$ and $k_2 = \frac{\delta t}{\tau_c + \delta t}$.

*Eq.* 3 integrates rotation $\Omega$ into tilt $G$. The variable $s$ acts as a switch: it is set to one during tilt and to 0 during EVAR (in which case $G$ remains equal to zero, independently of $\Omega$).

Finally, *Eq.* 4 that describes linear acceleration, resembles *Eq.* 1 in form and properties.

The system of these equations is rewritten as follows in order to eliminate $\Omega(t)$ from the right-hand side (which is needed so that it may fit into the form of *eq.* 7 below):

$$\begin{cases} \Omega(t) &= \quad\quad 0 \quad\quad + \quad \Omega^u(t) \quad + \quad \Omega^\varepsilon(t) \quad (eq.\,1b) \\ C(t) &= k_1.C(t-\delta t) \; + \quad k_2.\Omega^u(t) \quad + \quad k_2.\Omega^\varepsilon(t) \quad (eq.\,2b) \\ G(t) &= \quad G(t-\delta t) \; + \quad s.\delta t.\Omega^u(t) \quad + \quad s.\delta t.\Omega^\varepsilon(t) \quad (eq.\,3b) \\ A(t) &= \quad\quad 0 \quad\quad + \quad A^u(t) \quad + \quad A^\varepsilon(t) \quad (eq.\,4b) \end{cases}$$

The model sensory transduction is:

$$\begin{cases} V(t) &= \Omega(t) - C(t) \; + \; V^\eta(t) \quad (eq.\,5) \\ F(t) &= G(t) + A(t) \; + \; F^\eta(t) \quad (eq.\,6) \end{cases}$$

*Eq.* 5 indicates that the semicircular canals encode rotation velocity, minus the dynamic component $C$; and *Eq.* 6 indicates that the otolith organs encode the sum of tilt and acceleration.

## Model of head motion in matrix form

The system of equations $(1b - 4b)$ can be rewritten in matrix form:

$$X(t) = D.X(t - \delta t) + M.X^u(t) + E.X^\varepsilon(t) \qquad (eq.\ 7)$$

with:

$$X = \begin{bmatrix} \Omega \\ C \\ G \\ A \end{bmatrix}, D = \begin{bmatrix} 0 & 0 & 0 & 0 \\ 0 & k_1 & 0 & 0 \\ 0 & 0 & 1 & 0 \\ 0 & 0 & 0 & 0 \end{bmatrix}, M = \begin{bmatrix} 1 & 0 \\ k_2 & 0 \\ s.\delta t & 0 \\ 0 & 1 \end{bmatrix}, X^u = \begin{bmatrix} \Omega^u \\ A^u \end{bmatrix}, E = M, X^\varepsilon = \begin{bmatrix} \Omega^\varepsilon \\ A^\varepsilon \end{bmatrix}$$

Similarly, the model of sensory transduction $(eq.\ 5 - 6)$ is rewritten as:

$$S(t) = T.X(t) + S^\eta(t) \qquad (eq.\ 8)$$

With:

$$S = \begin{bmatrix} V \\ F \end{bmatrix}, T = \begin{bmatrix} 1 & -1 & 0 & 0 \\ 0 & 0 & 1 & 1 \end{bmatrix}, S^\eta = \begin{bmatrix} V^\eta \\ F^\eta \end{bmatrix}$$

Given the standard deviations of $\Omega^\varepsilon$, $A^\varepsilon$, $V^\eta$ and $F^\eta$ ($\sigma_\Omega$, $\sigma_A$, $\sigma_V$ and $\sigma_F$), the covariance matrices of $X^\varepsilon$ and $S^\eta$ (that are needed to perform Kalman filter computations) are respectively:

$$Q = E. \begin{bmatrix} \sigma_\Omega^2 & 0 \\ 0 & \sigma_A^2 \end{bmatrix}.E' \text{ and } R = \begin{bmatrix} \sigma_V^2 & 0 \\ 0 & \sigma_F^2 \end{bmatrix}$$

Note that the matrix $Q$ used here that unexpected rotations $\Omega^\varepsilon$ and accelerations $A^\varepsilon$ are independent. However, it could easily be adapted to represent more complex covariance matrices resulting for instance from motor noise.

## Kalman filter algorithm

The Kalman filter algorithm (***Kalman, 1960***) computes optimal state estimates in any model that follows the structure of $(eqn.\ 7, 8)$ (**Figure 1—figure supplement 2**). The optimal estimate $\hat{X}(t)$ is computed by the following steps (**Figure 1—figure supplement 2B**):

$$\begin{aligned} \hat{X}^p(t) &= D.\hat{X}(t - \delta t) + M.X^u(t) & \text{(state prediction)} \\ \hat{S}^p(t) &= T.\hat{X}^p(t) & \text{(predicted sensory signals)} \\ \delta S(t) &= S(t) - \hat{S}^p(t) & \text{(sensory errors)} \\ \hat{X}(t) &= \hat{X}^p(t) + K(t).\delta S(t) & \text{(final estimates)} \end{aligned}$$

The Kalman gain matrix $K(t)$ is computed as:

$$K(t) = L^p(t).T'.\left(T.L^p(t).T' + R\right)^{-1}$$

where $L^p(t) = D.L(t - \delta t).D' + Q$ and $L(t) = (Id - K(t).T).L^p(t)$ are the covariance of the predicted and updated estimates, $Q$ and $R$ are the covariance matrices of $E.X^\varepsilon$ and $S^\eta$, and $Id$ is an identity matrix. These equations are not shown in **Figure 1—figure supplement 2**.

The initial conditions of $\hat{X}$ are set according to the initial head position in the simulated motion, and the initial value of $K$ and $L$ are set to their steady-state value, which are computed by setting $L = Q$ and then running 500 iterations of the Kalman filter algorithm.

## Kalman filter algorithm developed

The inference is performed by applying the Kalman filter algorithm on *Eqs.* $7 - 8$. The corresponding computations can be expanded in the following equations:

## State predictions

$$\hat{\Omega}^p(t) = \Omega^u(t) \qquad (eq.\,1c)$$
$$\hat{C}^p(t) = k_1.\hat{C}(t-\delta t) + k_2.\hat{\Omega}^p(t) \quad (eq.\,2c)$$
$$\hat{G}^p(t) = \hat{G}(t-\delta t) + dt.\hat{\Omega}^p(t) \qquad (eq.\,3c)$$
$$\hat{A}^p(t) = A^u(t) \qquad (eq.\,4c)$$

## Sensory predictions

$$\hat{V}^p(t) = \hat{\Omega}^p(t) - \hat{C}^p(t) \quad (eq.\,5c)$$
$$\hat{F}^p(t) = \hat{G}^p(t) + \hat{A}^p(t) \quad (eq.\,6c)$$

## Sensory errors

$$\delta V(t) = V(t) - \hat{V}^p(t) \quad (eq.\,5d)$$
$$\delta F(t) = F(t) - \hat{F}^p(t) \quad (eq.\,6d)$$

## Final estimates

$$\hat{\Omega}(t) = \hat{\Omega}^p(t) + \Omega^k = \hat{\Omega}^p(t) + k^{\Omega}_{\delta V}.\delta V(t) + k^{\Omega}_{\delta F}.\delta F(t) \quad (eq.\,1d)$$
$$\hat{C}(t) = \hat{C}^p(t) + C^k = \hat{C}^p(t) + k^{C}_{\delta V}.\delta V(t) + k^{C}_{\delta F}.\delta F(t) \quad (eq.\,2d)$$
$$\hat{G}(t) = \hat{G}^p(t) + G^k = \hat{G}^p(t) + k^{G}_{\delta V}.\delta V(t) + k^{G}_{\delta F}.\delta F(t) \quad (eq.\,3d)$$
$$\hat{A}(t) = \hat{A}^p(t) + A^k = \hat{A}^p(t) + k^{A}_{\delta V}.\delta V(t) + k^{A}_{\delta F}.\delta F(t) \quad (eq.\,4d)$$

These equations form the basis of the model (in **Figure 9**, $k^{A}_{\delta V}$ and $k^{\Omega}_{\delta F}$ are assumed to be zero, see 'Kalman feedback gains' and **Table 2**).

## Velocity storage during EVAR

Here we analyze the Kalman filter equations to derive the dynamics of rotation perception during passive EVAR and compare it to existing models. During passive EVAR ($\hat{\Omega}^p = \Omega^u = 0$ and $\delta F = 0$), the dynamics of the rotation estimate depends of $\hat{C}$, which is governed by (eq. 2c, d):

$$\hat{C}^p(t) = k_1.\hat{C}(t-\delta t) \qquad (from\,eq.\,2c)$$
$$\hat{C}(t) = \hat{C}^p(t) + k^{C}_{\delta V}.\delta V \qquad (from\,eq.\,2d)$$

With:

$$\delta V = V(t) - \hat{V}^p(t) = V(t) + \hat{C}^p(t) \quad (from\,eq.\,5c,d)$$

Based on these equations, $\hat{C}$ follows a first-order differential equation:

$$\hat{C}(t) = \hat{C}^p(t) + k^{C}_{\delta V}.\left(V(t) + \hat{C}^p(t)\right) = k_1.\left(1 + k^{C}_{\delta V}\right).\hat{C}(t-\delta t) + k^{C}_{\delta V}.V(t) \quad (eq.\,2e)$$

This equation is characteristic of a leaky integrator, that integrates $V$ with a gain $k^{C}_{\delta V}$, and has a time constant $\tau_{VS}$ which is computed by solving:

$$(1 - \delta t/\tau_{VS}) = k_1.\left(1 + k^{C}_{\delta V}\right)$$

Based on the values of **Table 2**, we compute $\tau_{VS}$=16.5 s (in agreement with the simulations in **Figure 4—figure supplement 2**).

The final rotation estimate is the sum of $\hat{C}$ and the canal signal:

$$\hat{\Omega}(t) = k_{\delta V}^{\Omega}.\delta V(t) = k_1.\hat{C}(t-\delta t) + k_{\delta V}^{\Omega}.V(t) \qquad (from\ eq.\ 1d)$$

These equations reproduce the standard model of (**Raphan et al., 1979**). Note that the gains $k_{\delta V}^C = 0.19\ \delta t$, $\tau_{VS}$=16.5 s, and $k_{\delta V}^{\Omega} = 0.94$ are similar to the values in (**Raphan et al., 1979**) and to model fits to experimental data in (**Laurens et al., 2011**). The dynamics of $\hat{C}$ ($eq.\ 2e$) can be observed in simulations, i.e. in **Figure 4—figure supplement 2B** where the leaky integrator is charged by vestibular signals $V$ at t = 0 to 10 s and t = 60 to 70 s; and subsequently discharges with a time constant $\tau_{VS}$=16.5 s. The discharge of the integrator is also observed in **Figure 4—figure supplement 2C** when t > 60 s **Figure 6—figure supplement 3C** when t > 120 s.

## Passive tilt

Here we provide additional mathematical analyses about motion estimation during passive tilt. During passive tilt ($\hat{\Omega}^p = \Omega^u = 0; F = G$), the internal estimate $\hat{G}$ follows:

$$\hat{G}^p(t) = \hat{G}(t-\delta t) \qquad\qquad (from\ eq.\ 3c)$$
$$\hat{G}(t) = \hat{G}^p(t) + G^k \text{ with } G^k = k_{\delta V}^G.\delta V(t) + k_{\delta F}^G.\delta F(t) \qquad (eq.\ 3d)$$

First, we note that ($eq.\ 3c, d$) combine into $\hat{G}(t) = \int_0^t G^k$. In other words, the tilt estimate during passive tilt is computed by integrating feedback signals $G^k$.

Also, to a first approximation, the gain $k_{\delta V}^G$ is close to $\delta t$, the canal error $\delta V$ encodes $\Omega$ and $\delta F$ is approximately null. In this case, $G^k \approx \delta t.\Omega$ and $\hat{G}(t) \approx \int_0^t \Omega.\delta t$. Therefore, during passive tilt (**Figure 5**, **Figure 5—figure supplement 1**), the internal model ($eq.\ 3c$) integrates tilt velocity signals that originate from the canals and are conveyed by feedback pathways.

Note, however, that the Kalman gain $k_{\delta V}^G$ is slightly lower than $\delta t$ ($k_{\delta V}^G \approx 0.9\delta t$; **Table 2**). Yet, the final tilt estimate remains accurate due to an additional feedback originating from $\delta F$ which can be analyzed as follows. Because $k_{\delta V}^G < \delta t$, the tilt estimate $\hat{G}$ lags behind $G$, resulting in a small otolith error $\delta F = G(t) - \hat{G}^p(t)$ that contributes to the feedback signal (via the term $k_{\delta F}^G.\delta F(t)$ in $eq.\ 3d$). The value of $\delta F$ stabilizes to a steady-state where $\hat{G}(t) - \hat{G}(t-\delta t) = G(t) - G(t-\delta t) = \Omega.\delta t$. Based on $eq.\ 3d$, we obtain:

$$\hat{G}(t) - \hat{G}(t-\delta t) = k_{\delta V}^G.\delta V(t) + k_{\delta F}^G.\delta F(t) = \Omega.\delta t,$$

with $\delta V = \Omega$, $\delta F = (\delta t - k_{\delta V}^G).\Omega / k_{\delta F}^G \approx 0.13\ \Omega$ (based on **Table 2**).

Thus, a feedback signal originating from the otolith error complements the canal error. This effect is nevertheless too small to be appreciated in **Figure 5**.

## Kalman feedback gains

Here we provide additional information about Kalman gains and we justify that some feedback signals are considered negligible.

First, we note that some values of the Kalman gain matrix (those involved in a temporal integration), include the parameter $\delta t$ (**Table 2**). This is readily explained by the following example. Consider, for instance, the gain of the vestibular feedback to the tilt estimate ($k_{\delta V}^G$). During passive tilt, the tilt estimate $\hat{G}(t)$ is tracked by the Kalman filter according to:

$$\hat{G}(t) = \hat{G}(t-\delta t) + k_{\delta V}^G.\delta V(t) \qquad (from\ eq.\ 3c, d,\ with\ \delta F \approx 0)$$

Since $\hat{G}$ is computed by integrating canal signals ($\delta V$) over time, we would expect the equation above to be approximately equal to the following:

$$\hat{G}(t) = \hat{G}(t-\delta t) + \delta t.\delta V(t)$$

Therefore, we expect that $k_{\delta V}^G \approx \delta t$. When simulations are performed with $\delta t = 0.01$s, we find indeed that $k_{\delta V}^G = 0.009 = 0.9\ \delta t$. Furthermore, if simulations are performed again, but with $t = 0.1$s, we find $k_{\delta V}^G = 0.09 = 0.9\ \delta t$. In other words, the Kalman gain computed by the filter is

scaled as a function of $\delta t$ in order to perform the operation of temporal integration (albeit with a gain of 0.9). For this reason, we write $k_{\delta V}^{G} = 0.9\ \delta t$ in **Table 2**, which is more informative than $k_{\delta V}^{G} = 0.009$. Similarly, other Kalman gain values corresponding to $\hat{G}$ and $\hat{C}$ (which is also computed by temporal integration of Kalman feedback) are shown as a function of $\delta t$ in **Table 2**.

Furthermore, the values of $C^k$ and $G^k$ are divided by $\delta t$ in all figures, for the same reason. If, for example, $\delta V = 1$, then the corresponding value and $G^k$ would be 0.009 ($k_{\delta V}^{G}.\delta V$). This value is correct (since $k_{\delta V}^{G} = 0.9\ \delta t$) but would cause $G^k$ to appear disproportionately small. In order to compensate for this, we plot $G^k/dt$ in the figures. The feedback $C^k$ is scaled in a similar manner. In contrast, neither $\Omega^k$ nor $A^k$ are scaled.

Note that the feedback gain $k_{\delta V}^{A}$ (from the canal error $\delta V$ to $\hat{A}$) is equal to $-k_{\delta V}^{G}$ (**Table 2**). This compensates for a part of the error $\delta F$ during tilt (see previous section), which generates an erroneous acceleration feedback $A^k$. This component has a negligible magnitude and is not discussed in the text or included in the model of **Figure 9**.

Note also that the Kalman filter gain $k_{\delta F}^{\Omega}$ is practically equal to zero (**Table 2**). In practice, this means that the otoliths affect rotation perception only through variable $\hat{C}$. Accordingly, otolith-generated rotation signals (e.g. **Figure 6—figure supplement 3C**, from t = 60 s to t = 120 s) exhibit low-pass dynamics.

Because $k_{\delta V}^{A}$ and $k_{\delta F}^{\Omega}$ are practically null and have no measurable effect on behavioral or neuronal responses, the corresponding feedback pathways are excluded from **Figure 9**.

## Time constant of the somatogravic effect

Here we analyze the dynamics of the somatogravic effect. During passive linear acceleration, the otolith error $\delta F$ determines the feedback $G^k$ that aligns $\hat{G}$ with $F$ and therefore minimizes the feedback. This process can be modeled as a low-pass filter based on the following equations:

$$\delta F(t) = F(t) - \hat{F}^p(t) = A(t) - \hat{G}(t - \delta t)\ (eq.\ 3c, 6c, 6d)$$

$$\hat{G}(t) = \hat{G}^p(t) + G^k = \hat{G}(t - \delta t) + k_{\delta F}^{G}.\delta F(t)\ (eq.\ 3c, 3d,\ neglecting\ \delta V)$$

Leading to:

$$\hat{G}(t) = \hat{G}(t - \delta t) + k_{\delta F}^{G}.\left(A(t) - \hat{G}(t - \delta t)\right))$$

This equation illustrates that $\hat{G}$ is a low-pass filter that converges towards $A$ with a time constant $\tau_S = \delta t/k_{\delta F}^{G}$=1.3 s (**Table 2**).

Note that the feedback from $\delta F$ to $\hat{C}$ adds, indirectly, a second component to the differential equation above, leading $\hat{G}$ to transiently overshoot $A$ in **Figure 6—figure supplement 1**. Nonetheless, describing the somatogravic effect as a first-order low-pass filter is accurate enough for practical purposes.

## Model of motor commands

In this model, we assume that the Kalman filter receives copies of motor commands that encode rotation velocity $\Omega^u$ and linear acceleration $A^u$. We assume that motor noise and external perturbations amount to two independent Gaussian processes sum up to generate the total unpredictable components $\Omega^\varepsilon$ and $A^\varepsilon$. It should be noted that internal model computations that underlie motor control, and in particular the transformation of muscle activity into $\Omega^u$ and $A^u$, requires a series of coordinate transformations that are not modelled here, and that these transformation may affect the covariance matrix of motor noise and therefore of $X^\varepsilon$.

We found that this simplistic description of motor inputs to the Kalman filter was adequate in this study for two reasons. First, the motor inputs $\Omega^u$ and $A^u$ appear only in the prediction stage of the Kalman filter (*eq.* 7). In all simulations presented in this study, we have observed that motor commands were transformed into accurate predictions of motor commands. We

reason that, if the model of motor commands was changed, it would still lead to accurate predictions of the self-generated motion component, as long as the motor inputs are unbiased and are sufficient to compute all motion variables, either directly or indirectly through the internal model. Under these hypotheses, simulation results would remain unchanged.

Regarding the covariance matrix of motor noise, we find that assuming that the unpredictable motion components $\Omega^\varepsilon$ and $A^\varepsilon$ are independent is sufficient to model the experimental studies considered here. Furthermore, we note that the Kalman filter could readily accommodate a more sophisticated noise covariance matrix, should it be necessary.

## Visual rotation signals

In *Figure 4—figure supplement 2C*, a visual sensory signal *Vis* was added to the model as in (*Laurens, 2006*; *Laurens and Droulez, 2008*) by simply assuming that it encodes rotation velocity:

$$Vis(t) = \Omega + Vis^\eta$$

Where $Vis^\eta$ is a Gaussian noise with standard deviation $\sigma_{Vis}$ = 0.12 rad/s (*Laurens and Droulez, 2008*).

This signal is incorporated into the matrices of the sensory model as follows:

$$S = \begin{bmatrix} V \\ F \\ Vis \end{bmatrix}, T = \begin{bmatrix} 1 & -1 & 0 & 0 \\ 0 & 0 & 1 & 1 \\ 1 & 0 & 0 & 0 \end{bmatrix}, S^\eta = \begin{bmatrix} V^\eta \\ F^\eta \\ Vis^\eta \end{bmatrix}, R = \begin{bmatrix} \sigma_V^2 & 0 & 0 \\ 0 & \sigma_F^2 & 0 \\ 0 & 0 & \sigma_{Vis}^2 \end{bmatrix}$$

The model of head motion and the matrix equations of the Kalman filter remain unchanged.

## Model of head and neck rotations

We created a variant of the Kalman filter, where trunk velocity in space and head velocity relative to the trunk are two independent variables $\Omega_{TS}$ and $\Omega_{HT}$. We assumed that head position relative to the trunk is sensed by neck proprioceptors. To model this, we added an additional variable $N$ (for 'neck') that encodes the position of the head relative to the trunk: $N = \int \Omega_{HT}.dt$. We also added a sensory modality $P$ that represents neck proprioception.

Total head velocity (which is not an explicit variable in the model but may be computed as $\Omega = \Omega_{TS} + \Omega_{HT}$) is sensed through the semicircular canals, which were modeled as previously.

The model of head and trunk motion is based on the following equations:

$$\begin{cases} \Omega_{TS}(t) &= & 0 & + & \Omega_{TS}^u(t) & + & \Omega_{TS}^u(t) & (eq.\,S1) \\ \Omega_{HT}(t) &= & 0 & + & \Omega_{HT}^u(t) & + & \Omega_{HT}^\varepsilon(t) & (eq.\,S2) \\ N(t) &= & N(t-\delta t) + \delta t.\Omega_{HT}(t) & + & 0 & + & 0 & (eq.\,S3) \\ C(t) &= & k_1.C(t-\delta t) + k_2.(\Omega_{TS}(t) + \Omega_{HT}(t)) & + & 0 & + & 0 & (eq.\,S4) \end{cases}$$

Note that (*eq. S1*) and (*eq. S2*), are analogous to ((*eq. 1*) in the main model and imply that $\Omega_{TS}$ and $\Omega_{HT}$ are the sum of self-generated ($\Omega_{TS}^u$, $\Omega_{HT}^u$) and unpredictable components ($\Omega_{TS}^\varepsilon$, $\Omega_{HT}^\varepsilon$). Eq. S3 encodes $N = \int \Omega_{HT}.dt$. The canal model (*eq. S4*) is identical as in (*eq. 2*), the input being the velocity of the head in space, i.e. $\Omega_{TS}(t) + \Omega_{HT}(t)$.

The sensory model includes the canal signal $V$ and a neck proprioceptive signal $P$ that encodes neck position:

$$\begin{cases} V(t) &= & \Omega_{TS}(t) + \Omega_{HT}(t) - C(t) & + & V^\eta(t) & (eq.\,S5) \\ P(t) &= & N(t) & + & P^\eta(t) & (eq.\,S6) \end{cases}$$

Note that (*eq. S5*) is identical to (*eq. 5*) in the main model, and that $P$ is subject to sensory noise $P^\eta$.

Similar to the main model, (*eq. S1 − S6*) are written in matrix form:

$$X(t) = D.X(t-\delta t) + M.X^u(t) + E.X^\varepsilon(t) \qquad (eq.\,S7)$$

with:

$$X = \begin{bmatrix} \Omega_{TS} \\ \Omega_{HT} \\ N \\ C \end{bmatrix}, D = \begin{bmatrix} 0 & 0 & 0 & 0 \\ 0 & 0 & 0 & 0 \\ 0 & 0 & 1 & 0 \\ 0 & 0 & 0 & k_1 \end{bmatrix}, M = \begin{bmatrix} 1 & 0 \\ 0 & 1 \\ \delta t & 0 \\ k_2 & k_2 \end{bmatrix}, X^u = \begin{bmatrix} \Omega_{TS}^u \\ \Omega_{HT}^u \end{bmatrix}, E = M, X^\varepsilon = \begin{bmatrix} \Omega_{TS}^\varepsilon \\ \Omega_{HT}^\varepsilon \end{bmatrix}$$

Similarly, the model of sensory transduction (*eq. S5 − S6*) is rewritten as:

$$S(t) = T.X(t) + S^\eta(t) \qquad (eq.\ S8)$$

with:

$$S = \begin{bmatrix} V \\ P \end{bmatrix}$$

Given the standard deviations of $\Omega_{TS}^\varepsilon$, $\Omega_{HT}^\varepsilon$, $V^\eta$ and $P^\eta$ ($\sigma_{TS}$, $\sigma_{HT}$, $\sigma_V$ and $\sigma_P$), the covariance matrices of $X^\varepsilon$ and $S^\eta$ are respectively:

$$Q = E.\begin{bmatrix} \sigma_{TS}^2 & 0 \\ 0 & \sigma_{HT}^2 \end{bmatrix}.E'$$

Simulations were performed using the Kalman filter algorithm, as in the main model.

## Feedback signals during neck movement

In *Figure 7—figure supplements 2* and *3*, we note that passive neck motion induces a proprioceptive feedback $\delta P$ that encodes neck velocity, although proprioceptive signals $P$ are assumed to encode neck position. This dynamic transformation is explained by considering that, during passive motion:

$$\hat{N}^P(t) = \hat{N}(t - \delta t)$$

$$\hat{N}(t) = \hat{N}^P(t) + N^k = \hat{N}(t - \delta t) + k_{\delta V}^N.\delta V + k_{\delta P}^N.\delta P$$

Because $k_{\delta V}^N \approx 0$ (*Table 3*), neck position is updated exclusively by its own proprioceptive feedback $\delta P$. Furthermore, the equation above is transformed into:

$$\hat{N}(t) - \hat{N}(t - \delta t) = k_{\delta P}^N.\delta P$$

In a steady-state, $\hat{N}(t) - \hat{N}(t - \delta t) = N(t) - N(t - \delta t) = \Omega_{HT}.\delta t$, leading to:
$\delta P = \Omega_{HT}.\delta t / k_{\delta P}^N$ that is $\delta P \approx \Omega_{HT}.\delta t$ (with $k_{\delta P}^N \approx 1$)

Thus, similar to the reasons already pointed out in section 'Kalman feedback gains', the feedback signal $\delta P$ is scaled by $1/\delta t$ in *Figure 7—figure supplement 1–3*. Also, because the gain $k_{\delta P}^N$ (which is close to 1, see *Table 3*) doesn't scale with $\delta t$, the feedback $N^k = k_{\delta P}^N.\delta P$ is also scaled by $1/\delta t$ in *Figure 7—figure supplement 1–3*.

Importantly, the equations above indicate that neck proprioception error should encode neck velocity even when the proprioceptive signals are assumed to encode neck position.

Next, we note that, during passive neck rotation, the estimate of head velocity relative to the trunk is determined by $\hat{\Omega}_{HT} = k_{\delta P}^{\Omega_{HT}}.\delta P$. Since $\delta P \approx \Omega_{HT}.\delta t$, we expect that $k_{\delta P}^{\Omega_{HT}} \approx 1/\delta t$. Accordingly, simulations performed with $\delta t = 0.01s$ yield $k_{\delta P}^{\Omega_{HT}} = 89 = 0.89/\delta t$. Furthermore, performing simulations with $\delta t = 0.1s$ leads to $k_{\delta P}^{\Omega_{HT}} = 8.9 = 0.89/\delta t$. We find that $k_{\delta P}^{\Omega_{TS}}$ is also dependent of $\delta t$. Therefore, the corresponding Kalman gains scale with $\delta t$ in *Table 3*.

Note that the considerations above can equally explain the amplitude and dynamics of the predicted neck position and of the neck proprioceptive error in *Figure 7—figure supplement 6*. The simulation in *Figure 7—figure supplement 6C* is identical (with half the amplitude) to a passive rotation of the head relative to the trunk (*Figure 7—figure supplement 2*), where $\delta P = \delta V > 0$. The simulation in *Figure 7—figure supplement 6C* can be explained mathematically by noticing that it is equivalent to an active head motion (where $\delta P = 0$) superimposed to a passive rotation of the head with a gain of 0.5 and in the opposite direction, resulting in $\delta P = \delta V < 0$.

# Three-dimensional Kalman filter

For the sake of simplicity, we have restricted our model to one dimension in this study. However, generalizing the model to three-dimensions may be useful for further studies and is relatively easily accomplished, by (1) replacing one-dimensional variables by three-dimensional vectors and (2) locally linearizing a non-linearity that arises from a vectorial cross-product (*Eq. 9* below), as shown in this section.

## Principle

Generalization of the Kalman filter to three dimensions requires replacing each motion and sensory parameter with a 3D vector. For instance, $\Omega$ is replaced by $\Omega_x$, $\Omega_y$ and $\Omega_z$ that encode the three-dimensional rotation vector in a head-fixed reference frame $(x, y, z)$. Sensory variables are also replaced by three variables, that is $(V_x, V_y, V_z)$ and $(F_x, F_y, F_z)$ that encode afferent signals from the canals and otoliths in three dimensions.

With one exception, all variables along one axis (e.g. $\Omega_x$, $C_x$, $G_x$, $A_x$, $V_x$, $F_x$ along the $x$ axis) are governed by the same set of equations (*eqn* $1-6$) as the main model. Therefore, the full 3D model can be thought of as three independent Kalman filters operating along the $x, y$ and $z$ dimensions. The only exception is the three-dimensional computation of tilt, which follows the equation:

$$\vec{G}(t) = \vec{G}(t-\delta t) + \delta t.\vec{G}(t-\delta t) \times \vec{\Omega}(t) \quad (eq. 9)$$

Where $\vec{G}$ and $\vec{\Omega}$ are vectorial representations of $(G_x, G_y, G_z)$ and $(\Omega_x, \Omega_y, \Omega_z)$, and $\times$ represents a vectorial cross-product. In matrix form,

$$\delta t.\vec{G} \times \vec{\Omega} = \delta t. \begin{bmatrix} G_x \\ G_y \\ G_z \end{bmatrix} \times \begin{bmatrix} \Omega_x \\ \Omega_y \\ \Omega_z \end{bmatrix} = \delta t. \begin{bmatrix} G_y.\Omega_z - G_z.\Omega_y \\ G_z.\Omega_x - G_x.\Omega_z \\ G_x.\Omega_y - G_y.\Omega_x \end{bmatrix}$$

This non-linearity is implemented by placing the terms $\delta t$ and $(G_x, G_y, G_z)$ in the matrices $M$ and $E$ that integrate rotation inputs ($\Omega^u$ and $\Omega^\varepsilon$) into tilt, as shown below.

## Implementation

The implementation of the 3D Kalman filter is best explained by demonstrating how the matrices of the 1D filter are replaced by scaled-up matrices.

First, we replace all motion variables and inputs by triplets of variables along $x, y$ and $z$:

$$X = \begin{bmatrix} \Omega \\ C \\ G \\ A \end{bmatrix}, X^u = \begin{bmatrix} \Omega^u \\ A^u \end{bmatrix}, X^\varepsilon = \begin{bmatrix} \Omega^\varepsilon \\ A^\varepsilon \end{bmatrix} \text{ are replaced by: } X = \begin{bmatrix} \Omega_x \\ \Omega_y \\ \Omega_z \\ C_x \\ C_y \\ C_z \\ G_x \\ G_y \\ G_z \\ A_x \\ A_y \\ A_z \end{bmatrix},$$

$$X^u = \begin{bmatrix} \Omega^u_x \\ \Omega^u_y \\ \Omega^u_z \\ A^u_x \\ A^u_y \\ A^u_z \end{bmatrix}, X^\varepsilon = \begin{bmatrix} \Omega^\varepsilon_x \\ \Omega^\varepsilon_y \\ \Omega^\varepsilon_z \\ A^\varepsilon_x \\ A^\varepsilon_y \\ A^\varepsilon_z \end{bmatrix}$$

Next, we scale the matrix $D$ up; each non-zero element in the 1D version being repeated twice in the 3D version:

$$D = \begin{bmatrix} 0 & 0 & 0 & 0 \\ 0 & k_1 & 0 & 0 \\ 0 & 0 & 1 & 0 \\ 0 & 0 & 0 & 0 \end{bmatrix} \text{ is replaced by } D = \begin{bmatrix} 0 & 0 & 0 & 0 & 0 & 0 & 0 & 0 & 0 & 0 & 0 & 0 \\ 0 & 0 & 0 & 0 & 0 & 0 & 0 & 0 & 0 & 0 & 0 & 0 \\ 0 & 0 & 0 & 0 & 0 & 0 & 0 & 0 & 0 & 0 & 0 & 0 \\ 0 & 0 & 0 & k_1 & 0 & 0 & 0 & 0 & 0 & 0 & 0 & 0 \\ 0 & 0 & 0 & 0 & k_1 & 0 & 0 & 0 & 0 & 0 & 0 & 0 \\ 0 & 0 & 0 & 0 & 0 & k_1 & 0 & 0 & 0 & 0 & 0 & 0 \\ 0 & 0 & 0 & 0 & 0 & 0 & 1 & 0 & 0 & 0 & 0 & 0 \\ 0 & 0 & 0 & 0 & 0 & 0 & 0 & 1 & 0 & 0 & 0 & 0 \\ 0 & 0 & 0 & 0 & 0 & 0 & 0 & 0 & 1 & 0 & 0 & 0 \\ 0 & 0 & 0 & 0 & 0 & 0 & 0 & 0 & 0 & 0 & 0 & 0 \\ 0 & 0 & 0 & 0 & 0 & 0 & 0 & 0 & 0 & 0 & 0 & 0 \\ 0 & 0 & 0 & 0 & 0 & 0 & 0 & 0 & 0 & 0 & 0 & 0 \end{bmatrix}$$

We build $M$ in a similar manner. Furthermore, the element $s.\delta t$ (that encodes the integration of $\Omega^u$ into $G$) is replaced by a set of terms that encode the cross-product in ($eq.\ 9$), as follows:

$$\delta t.\vec{G} \times \vec{\Omega} = \delta t. \begin{bmatrix} G_x \\ G_y \\ G_z \end{bmatrix} \times \begin{bmatrix} \Omega_x \\ \Omega_y \\ \Omega_z \end{bmatrix} = \delta t. \begin{bmatrix} G_y.\Omega_z - G_z.\Omega_y \\ G_z.\Omega_x - G_x.\Omega_z \\ G_x.\Omega_y - G_y.\Omega_x \end{bmatrix} = \delta t. \begin{bmatrix} 0 & -G_z & G_y \\ G_z & 0 & -G_x \\ -G_y & G_x & 0 \end{bmatrix} \times \begin{bmatrix} \Omega_x \\ \Omega_y \\ \Omega_z \end{bmatrix}$$

$$M = \begin{bmatrix} 1 & 0 \\ k_2 & 0 \\ s.\delta t & 0 \\ 0 & 1 \end{bmatrix} \text{ is replaced by } M = \begin{bmatrix} 1 & 0 & 0 & 0 & 0 & 0 \\ 0 & 1 & 0 & 0 & 0 & 0 \\ 0 & 0 & 1 & 0 & 0 & 0 \\ k_2 & 0 & 0 & 0 & 0 & 0 \\ 0 & k_2 & 0 & 0 & 0 & 0 \\ 0 & 0 & k_2 & 0 & 0 & 0 \\ 0 & -G_z.\delta t & G_y.\delta t & 0 & 0 & 0 \\ G_z.\delta t & 0 & -G_x.\delta t & 0 & 0 & 0 \\ -G_y.\delta t & G_x.\delta t & 0 & 0 & 0 & 0 \\ 0 & 0 & 0 & 1 & 0 & 0 \\ 0 & 0 & 0 & 0 & 1 & 0 \\ 0 & 0 & 0 & 0 & 0 & 1 \end{bmatrix}$$

As previously, $E = M$. Note that $M$ and $E$ must be recomputed at each iteration since $G_x$, $G_y$ and $G_z$ change continuously.

Similarly, we build the sensor model as follows:

$$S = \begin{bmatrix} V \\ F \end{bmatrix} \text{ and } S^\eta = \begin{bmatrix} V^\eta \\ F^\eta \end{bmatrix} \text{ are replaced by } S = \begin{bmatrix} V_x \\ V_y \\ V_z \\ F_x \\ F_y \\ F_z \end{bmatrix} \text{ and } S^\eta = \begin{bmatrix} V_x^\eta \\ V_y^\eta \\ V_z^\eta \\ F_x^\eta \\ F_y^\eta \\ F_z^\eta \end{bmatrix}$$

$$T = \begin{bmatrix} 1 & -1 & 0 & 0 \\ 0 & 0 & 1 & 1 \end{bmatrix} \text{ is replaced by } T = \begin{bmatrix} 1 & 0 & 0 & -1 & 0 & 0 & 0 & 0 & 0 & 0 & 0 & 0 \\ 0 & 1 & 0 & 0 & -1 & 0 & 0 & 0 & 0 & 0 & 0 & 0 \\ 0 & 0 & 1 & 0 & 0 & -1 & 0 & 0 & 0 & 0 & 0 & 0 \\ 0 & 0 & 0 & 0 & 0 & 0 & 1 & 0 & 0 & 1 & 0 & 0 \\ 0 & 0 & 0 & 0 & 0 & 0 & 0 & 1 & 0 & 0 & 1 & 0 \\ 0 & 0 & 0 & 0 & 0 & 0 & 0 & 0 & 1 & 0 & 0 & 1 \end{bmatrix}$$

$$Q = E. \begin{bmatrix} \sigma_\Omega^2 & 0 \\ 0 & \sigma_A^2 \end{bmatrix}.E' \text{ and } R = \begin{bmatrix} \sigma_V^2 & 0 \\ 0 & \sigma_F^2 \end{bmatrix} \text{ are replaced by:}$$

$$Q = E. \begin{bmatrix} \sigma_\Omega^2 & 0 & 0 & 0 & 0 & 0 \\ 0 & \sigma_\Omega^2 & 0 & 0 & 0 & 0 \\ 0 & 0 & \sigma_\Omega^2 & 0 & 0 & 0 \\ 0 & 0 & 0 & \sigma_A^2 & 0 & 0 \\ 0 & 0 & 0 & 0 & \sigma_A^2 & 0 \\ 0 & 0 & 0 & 0 & 0 & \sigma_A^2 \end{bmatrix}.E' \text{ and } R = \begin{bmatrix} \sigma_V^2 & 0 & 0 & 0 & 0 & 0 \\ 0 & \sigma_V^2 & 0 & 0 & 0 & 0 \\ 0 & 0 & \sigma_V^2 & 0 & 0 & 0 \\ 0 & 0 & 0 & \sigma_F^2 & 0 & 0 \\ 0 & 0 & 0 & 0 & \sigma_F^2 & 0 \\ 0 & 0 & 0 & 0 & 0 & \sigma_F^2 \end{bmatrix}$$

