## [Decision Letter]

Thank you for submitting your article "A unified internal model theory to resolve the paradox of active versus passive self-motion sensation" for consideration by *eLife*. Your article has been reviewed by three peer reviewers, and the evaluation has been overseen by a Senior Editor and a Reviewing Editor. The following individual involved in review of your submission has agreed to reveal his identity: Laurence Harris (Reviewer #3).

The reviewers have discussed the reviews with one another and the Reviewing Editor has drafted this decision to help you prepare a revised submission.

Summary:

This very timely and comprehensive theory paper studies the way self-motion may be estimated by the brain and mainly concerns the vestibular system during active and passive head movements. In particular, it suggests a unified theory for how active and passive motion can be estimated by a single internal model. This extends previous, well-established work by the authors and others applying internal models to the problem of estimation of passive motion. A Kalman filter framework was used to determine optimal (Bayesian) solutions. The paper draws on a considerable amount of previously published experimental evidence from a rich series of elegant experiments to argue the hypothesis and to consolidate a large amount of previously collected data into an explanatory model. The internal forward model proposed here is used to make a series of specific experimental predictions (mainly found in the supplementary information). The model's mathematical development and execution is very thoroughly motivated, as it draws on a solid body of previous computational work. The manuscript did an excellent job of simplifying the description of such complex models. Model simulation results are shown for different scenarios and compared descriptively with neurophysiologic evidence. A particularly strong point about the paper is that it makes multiple testable predictions and could be used, for example, to predict the consequences of experience in microgravity. The principle of this type of model may find application beyond the situation considered here, for example, in looking at how the sensory consequences of an eye movement are combined with retinal information to provide the perception of an apparently stable world (e.g., Bridgeman et al. 1994).

The reviewers appreciate the structure of the paper and the systematic testing using different conditions building in complexity. The unusual interactive connection with the supplemental material to try and keep the main paper shorter is also appreciated.

Essential revisions:

1) Novelty: A major concern for the paper is the nature of the advancement of our understanding of computational principles in the brain: Internal forward models, priors and their impact on error-based movement correction have been extensively studied in the human reaching / motor control literature (from Wolpert et al. to Bernicker et al.; some of this should be mentioned to make a connection to this body of literature) and as such their ability to explain active and passive motion (of arms) are not especially novel. Thus, the main advance must lie in the domain of vestibular information processing, which needs to motivated better.

2) Cancellation mechanism: The authors state that "we have tested the hypothesis that this postulated cancellation mechanism uses exactly the same sensory internal model computations…." While the authors have put forward one framework that explains a variety of phenomenon, the reviewers wonder if other models could also explain these phenomena. Perhaps (at least for some of the scenarios) the motor command could be subtracted from the sensory estimate of a "passive only" internal model and yield the same response. This should be addressed by a comparison to an alternative "subtraction model". The simple model is probably what people naively think might be the case and so demonstrating its failings might give the paper more impact.

3) Reliance of motor command generation: The model does not include an additional feedback pathway – the reliance of motor command generation on sensory estimates. For example, a passive head movement could result in a stabilizing active motor command. Or an active head movement could be less than desired because of noise, requiring an adjustment of the motor command to compensate. Obviously these additional pathways complicate the solution to the Kalman filter gains. While the model is still an important contribution without these pathways, the limitations of the model without these pathways should be addressed.

4) Motor commands: The authors assume that the motor command is a noiseless signal that is perfectly executed. This apparently doesn't account for motor error, since the effect of motor commands is not deterministic and noiseless, as assumed by injecting the motor command as Xu (see Figure 1) into the system model. It seems that this ignores motor noise and execution noise, even though these sources of noise are extremely important (see work by Wolpert and others). In the second paragraph of the subsection “Model of motor commands” the authors claim that 𝑋^𝜀^ includes motor noise, but this is not evident and needs to be discussed better. Motor noise would, in this reviewer's opinion, enter before applying M and therefore would have a different noise spectrum and/or covariance matrix than the system noise 𝑋^𝜀^.

An alternative view would be that processed efference copy signals (i.e. predicted sensory signals) have to be treated the same way as other sensory signals, because 1) the efference copy just like any neural signal cannot be assumed to be noiseless and 2) it would take into account the motor noise appropriately (instead of treating it as external perturbation). Is that view wrong? Or would it provide an alternative to your present model? It seems to me that approaches such as described in papers by Wolpert et al. would consider efference copy input as containing noise. For example, MacNeilage and Glasauer (Front Comput Neurosci 2017) assume that vestibular and efference copy signals are combined like two sensory signals, weighted by their reliability.

5) Reference frames: a theoretical concern is how do the motor signals (and proprioceptive signals) arrive in the correct form and frame of reference to be compared to the sensory information?

6) Active vs. passive movements: As mentioned in the text (subsection “Interruption of internal model computations during proprioceptive mismatch”), the current model cannot reproduce some important results concerning active movements: when active head movement is blocked or torque is applied, "central neurons were shown to encode net head motion". The authors suggest that this "result may be reproduced by the Kalman filter by switching off the internal model of the motor plant.…". However, this actually means that the Kalman filter model fails to explain this result, because it would need another element. What the authors propose here sounds like an ad-hoc extension of the model (a switch) to explain results, which can as well be interpreted as evidence against the model. It is important to discuss this more appropriately.

In the second paragraph of the aforementioned subsection, the authors suggest that "proprioceptive mismatch" would cause to stop using the internal model. However, that seems a bit oversimplifying: Proprioceptive mismatch occurs due to a variety of natural conditions (e.g. increased head inertia due to carrying food, or due to fatigue, etc.) for which it would make no sense at all to completely stop using the internal model.

7) Parameter variations: Technically, systematic parameter variations to test for the robustness of the model predictions are missing (which can be easily addressed).

8) One reviewer would also like to see more quantitative results showing time courses of adaptation in the model and e.g. firing rates match up with each other. This brings up the question of whether the model does adapt or whether all Kalman filters operate with steady state gain (as suggested by another reviewer).

---

## [Author Response]

Essential revisions:1) Novelty: A major concern for the paper is the nature of the advancement of our understanding of computational principles in the brain: Internal forward models, priors and their impact on error-based movement correction have been extensively studied in the human reaching / motor control literature (from Wolpert et al. to Bernicker et al.; some of this should be mentioned to make a connection to this body of literature) and as such their ability to explain active and passive motion (of arms) are not especially novel. Thus, the main advance must lie in the domain of vestibular information processing, which needs to motivated better.

We agree with the reviewers that the general framework of the internal model has been largely validated by studies of human motor control and has been used to build quantitative dynamic models of trajectory planning (e.g. work on optimal control by Todorov; studies of saccade dynamics by Chen-Harris and eye-head gaze shifts by Saglam and Glasauer) or adaptation (e.g. Kalman filter model by Berlinker and Körding). We added references to this literature in the Introduction; and discussed it in “Discussion: Relation to previous dynamical models” to highlight this connection. However, our model is the only one, to our knowledge, to make the link between sensory dynamics (i.e. the canals), ambiguities (i.e. the otoliths), priors and motor efference copies to simulate the time course of sensory illusions during passive motion.

Perhaps most importantly, the focus of this study is to model neuronal responses – and this is what our simulations deal with. Thus, it is very different from studies of motor control, where the complexity of the system prevents easy prediction about neural responses. By simulating and explaining neuronal responses during active and passive self-motion in the light of a quantitative model (already well-established during passive motion), this study advances our understanding of how theoretical principles map onto brainstem and cerebellar circuits, and will guide further experimental testing in the future. In our view, this is an advance that extends far beyond the field of vestibular processing.

Furthermore, understanding how efference copies contribute to self-motion perception will impact many fields connected to the vestibular field, such as that of balance and locomotor control. We discuss in the manuscript how this model clarifies the role of vestibular signals in locomotion and explains the consequences of vestibular deficits. As stated in Berniker and Körding, 2011: “A crucial first step for motor control is therefore to integrate sensory information reliably and accurately”, and practically any locomotor activity beyond reaching movements in seated subjects will affect posture and therefore recruit the vestibular sensory modality. This study will also impact the field of navigation: there is currently a debate why head direction cell tuning disappears in vestibular deficient animals. These papers reason that, since vestibular signals are cancelled in actively moving animals, removing the vestibular organs should not make a difference, which is exactly the sort of misconception that our study aims at dissipating. By providing a state of the art model of self-motion processing during active and passive motion, we are bridging several noticeable gaps between the vestibular and motor control/navigation fields.

We would also like to point out that the internal model framework presented here accounts for and explains more than half a century of behavioral and neurophysiological experimental findings under a wide range of conditions (active versus passive self-motion, visuo-vestibular interactions; canal-otolith interactions; head/neck rotations). This demonstration of the power of the internal model framework in accounting for and predicting neural responses should rejoice the motor control community, where neurophysiological testing of this framework has been limited.

We have redesigned Figure 1 to better summarize the relation between our work and previous studies and to illustrate (panel D) how the Kalman filter computations presented here may relate to and interact with internal model computations that underlie motor control. The original Figure 1 has been moved to supplementary figures (Figure 1—figure supplement 2).

2) Cancellation mechanism: The authors state that "we have tested the hypothesis that this postulated cancellation mechanism uses exactly the same sensory internal model computations…" While the authors have put forward one framework that explains a variety of phenomenon, the reviewers wonder if other models could also explain these phenomena. Perhaps (at least for some of the scenarios) the motor command could be subtracted from the sensory estimate of a "passive only" internal model and yield the same response. This should be addressed by a comparison to an alternative "subtraction model". The simple model is probably what people naively think might be the case and so demonstrating its failings might give the paper more impact.

We are now considering this and other alternative models in Figure 1—figure supplement 1 (and Figure 9) and we have added a section at the end of the Results to present the alternative models. The alternative model suggested by the reviewers can reproduce the suppression of neuronal responses during short-duration rotation, but produces different results from the Kalman filter during translation or long-duration rotations. The idea is that, during brief rotations, both the motor command and the “passive” model would produce estimates that are close to the real motion, and therefore one may cancel another. In contrast, it is well known that passive long-duration rotations, as well as translations, induce motion illusions. When these movements are performed actively, the output of the “passive” model would be affected by these illusions whereas the motor command would encode real motion and therefore they would not cancel each other (see simulations in Figure 9).

Can we refute this alternative model based on experimental evidence? To date, we are not aware of neuronal recording studies that has experimentally tested these predictions. Testing this alternative “subtraction model” would require recording VN or rFN neurons during long-duration active rotations; or to record them during tilt and translations with sufficient experimental control to test whether they are affected by the somatogravic effect (as in Figure 9). To our knowledge, this has not been done.

But we can refute the “subtraction” model based on behavioral evidence. Specifically, during long-duration active rotation, rotation perception and the VOR persist indefinitely (although it saturates at about 10°/s in humans, unlike in monkeys; Solomon and Cohen 1992; humans: Guedry and Benson 1983). As shown in Figure 9, the subtraction model cannot reproduce this.

Furthermore, post-rotatory responses (i.e. the “negative” rotation estimate that occurs when the rotation stops) are reduced following active rotations in humans and completely cancelled in monkeys (Solomon and Cohen 1992; humans: Guedry and Benson 1983, Howard et al. 1995). These experimental findings severely contradict the “subtraction” model, because the motor command would stop when the motion stops, and the “passive” model would output the same post-rotatory response as during passive motion. Therefore, the “subtraction” model would predict the same post-rotatory response during active and passive motion. In contrast, the Kalman filter predicts that the motor commands are fed through an internal model of the canals (i.e. the same internal model that generates a “velocity storage” signal during passive motion), and this internal model anticipates and cancels post-rotatory responses.

Another way to understand why the subtraction model would not work is to consider the question of how the brain would generate the correct final self-motion estimate using the difference between the sensory estimate and the motor command as a correction signal. That is, (final estimate) = (motor prediction) + (correction signal). However, if the motor command is identical to the prediction (based on the subtraction model), this would develop as (final estimate) = (motor prediction) + (sensory estimate-motor prediction) = sensory estimate. In other words, this model would predict that the final self-motion estimate is driven by vestibular signals only, even during active motion, in contradiction with the behavioral findings that show an improvement of self-motion perception during active motion. We also consider in Figure 9 other possibilities where the final estimate would be a weighted average of the motor command and sensory predictions, and we demonstrate that none of these combinations accounts for experimental findings (such as the reduction of post-rotatory responses following active rotation).

We thank the reviewers for bringing this up, as it has helped make our argument for the optimal (Kalman) model stronger.

3) Reliance of motor command generation: The model does not include an additional feedback pathway – the reliance of motor command generation on sensory estimates. For example, a passive head movement could result in a stabilizing active motor command. Or an active head movement could be less than desired because of noise, requiring an adjustment of the motor command to compensate. Obviously these additional pathways complicate the solution to the Kalman filter gains. While the model is still an important contribution without these pathways, the limitations of the model without these pathways should be addressed.

Indeed we have not modelled this feedback pathway because we did not want to complicate an already complex story. A previous Kalman filter model that focused on postural control has done it (van der Kooij et al. 2001). We now show this up front (Figure 1) and refer to this again in the Discussion: “Further behavioral evidence” and mention the possibility of extending the Kalman filter approach to model neuronal pathways involved in head stabilization (i.e. the vestibulo-collic reflex and postural control). We agree that this would be a very interesting topic for future work.

4) Motor commands: The authors assume that the motor command is a noiseless signal that is perfectly executed. This apparently doesn't account for motor error, since the effect of motor commands is not deterministic and noiseless, as assumed by injecting the motor command as Xu (see Figure 1) into the system model. It seems that this ignores motor noise and execution noise, even though these sources of noise are extremely important (see work by Wolpert and others). In the second paragraph of the subsection “Model of motor commands” the authors claim that 𝑋^𝜀^ includes motor noise, but this is not evident and needs to be discussed better. Motor noise would, in this reviewer's opinion, enter before applying M and therefore would have a different noise spectrum and/or covariance matrix than the system noise 𝑋^𝜀^.

We apologize for not clarifying this point better. We use 𝑋^𝜀^ to model the responses to passive vestibular stimulation through most sections of the manuscript. However, the model does *not* assume that the motor command is a noiseless signal. Instead, it assumes that motor errors and externally generated perturbations are Gaussian and additive. The sum of multivariate Gaussians is a multivariate Gaussian and the term E. 𝑋^𝜀^ represents this sum. Note that this approach can generally be applied even when motor noise and external perturbations have different covariance matrices.

Here we have used a simple assumption that Ω^ε^ (the sum of ‘rotational’ motor errors and passive rotations) and A^ε^ (the sum of ‘translational’ motor errors and passive linear accelerations) are independent. Motor noise could indeed have a covariance matrix where these errors are not independent, and this could be easily introduced in the model by a simple adjustment of the matrix Q. Therefore, our choice to assume a simple covariance matrix should not be seen as weakness but instead of the application of the principle of parsimony. Our aim is to model the largest amount of experimental data (within the scope of the study) with the minimal number of parameters, and there are no data (in the experimental studies considered here) that require the adoption of a more complex covariance matrix. Should any subsequent study focus on this issue, the Kalman filter may easily be adapted for this purpose. We discuss this point in Supplementary Methods: “Model of motor commands”.

An alternative view would be that processed efference copy signals (i.e. predicted sensory signals) have to be treated the same way as other sensory signals, because 1) the efference copy just like any neural signal cannot be assumed to be noiseless and 2) it would take into account the motor noise appropriately (instead of treating it as external perturbation). Is that view wrong? Or would it provide an alternative to your present model? It seems to me that approaches such as described in papers by Wolpert et al. would consider efference copy input as containing noise. For example, MacNeilage and Glasauer (Front Comput Neurosci 2017) assume that vestibular and efference copy signals are combined like two sensory signals, weighted by their reliability.

In our model, treating efference copies as sensory signals (and introducing them in the sensory signal matrix S) would change the model’s results. This relates to the way the motor command and the prior on external motion are combined, as can be explained by the following example. Consider a short rotation (ignoring the dynamics of the canals), and suppose that the vestibular signal has a noise of σv = 10°/s, the motor noise has a standard deviation σm = 10°/s and the prior on external perturbations a standard deviation σp = 39°/s (together, σm and σp combine to form the total unpredictable component σε = sqrt(σm^2 + σp^2)≈40°/s). Suppose further that the motor commands encodes a rotation at 100°/s but the canals sense a velocity of 0°/s.

In our model, the external perturbations are added to the motor command to form a predictive signal with an average of 100°/s and a standard deviation of 40°/s. This will be combined with the signals from the canals (average of 0° and standard deviation of 10°/s) according to the rules of sensory fusion, resulting in a weight of 0.94 for the canals (identical to the vestibular gain k^Omega_deltaV in Table 2), 0.06 for the motor command and a final estimate of 6°/s.

If the motor commands were treated as sensory signals (like the vestibular signals), then the model would make a fusion between 3 Gaussians: the prior on external perturbations (average: 0°/s, standard deviation: 39°/s), the vestibular signals (average: 0°/s, standard deviation: 10°/s) and the motor command (average: 100°/s, standard deviation: 10°/s). The motor command and vestibular signals, having identical levels of noise, would receive identical weights (of 0.48) and the final estimate would be 48°/s. As a conclusion, the basic assumption in sensory fusion is that the reading of a sensor is equal to the variable of interest, plus sensory noise. In the case of motor commands, the efference copy is equal to head motion, plus motor noise, plus external motion. Therefore, treating the motor command as a sensory modality is possible only if the “sensory weight” of the motor commands reflects both the motor noise and the distribution of external motion.

Another aspect of this question is how Kalman filter relate to weighted averaging. It is indeed possible to approximate the Kalman filter model using weighted averaging operations, assuming (1) short duration stimuli where the internal model’s dynamics can be neglected and (2) that “noise” added to efference copies encompasses external perturbations as well as motor noise (which is what MacNeilage and Glasauer did; c.f. the fourth paragraph of the Introduction). Therefore, their study uses similar concepts to our model – and we thank the reviewer for bringing it to our attention: we now refer to it in the Discussion: “Further behavioral evidence”.

5) Reference frames: a theoretical concern is how do the motor signals (and proprioceptive signals) arrive in the correct form and frame of reference to be compared to the sensory information?

This is indeed a theoretical necessity, and we discuss this point in Supplementary Methods: “Model of motor commands”. The question of how the brain performs the reference frame transformations necessary for motor planning and proprioception is certainly a very exciting and challenging question for the motor control community. While we would not presume to model the neurophysiological operations that underlie these multiple reference frame transformations, we would point out that the present and previous publications address this question to a certain extent. Indeed, the transformation of rotation signals onto tilt derivative signals (represented by “3D” boxes in Figure 10) and its integration over time amounts to a 3D coordinate transformation (from egocentric to a gravity-centered reference frame) and requires the brain to overcome the non-commutativity of rotations. One of our previous studies (Laurens and Angelaki, Neuron 2013) demonstrated that cerebellar circuits were able to perform this coordinate transformation (i.e. identified the tilt-selective PC represented in Figure 10 and demonstrated that they convert 3D rotation signals into a tilt signal referenced to gravity). Thus, our model actually encompasses one 3D coordinate transformation whose neuronal correlate has been identified experimentally.

By the way, this point illustrates the fact that, while the vestibular field is considerably simpler than the motor control field, it shares many common questions such as that of 3D coordinate transformations and dynamic Bayesian inference. Furthermore, the vestibular field benefits from the fact that it can be modelled and studied using relatively few variables; it can be supported by a very powerful theoretical framework and very powerful quantitative models, and a sizeable fraction of neuronal populations in vestibular areas of the brainstem and cerebellum have been characterized. We would like to think that it could serve as a valuable model system for investigating the neuronal implementation of computational principles (reference frame transformations, dynamics Bayesian inference…) that are relevant to the motor control community.

6) Active vs. passive movements: As mentioned in the text (subsection “Interruption of internal model computations during proprioceptive mismatch”), the current model cannot reproduce some important results concerning active movements: when active head movement is blocked or torque is applied, "central neurons were shown to encode net head motion". The authors suggest that this "result may be reproduced by the Kalman filter by switching off the internal model of the motor plant.…". However, this actually means that the Kalman filter model fails to explain this result, because it would need another element. What the authors propose here sounds like an ad-hoc extension of the model (a switch) to explain results, which can as well be interpreted as evidence against the model. It is important to discuss this more appropriately.

We thank the reviewers for motivating us to discuss this point better. We are now introducing this question in the Introduction, and we discuss it in more detail.

In brief, several experiments from the Cullen lab demonstrated that, whenever proprioceptive errors occur during active neck movement, neuronal responses become non-linear in the sense that neurons encode the total head motion irrespective of the actual motor command. It is obvious that this cannot be modelled by a Kalman filter (which is linear) and therefore simulating these properties requires an extension of the model. We added a figure (Figure 7—figure supplement 7) that summarizes these findings and points out this non-linearity.

We do not see this as a weakness of the model. The whole point of the Kalman filter is to model the (linear) internal model computations that has been shown experimentally to affect neuronal responses (linearly) in a wide range of conditions, including combination of active motion and passive whole body rotations; and the Kalman filter does it well. Cullen’s results have demonstrated that proprioceptive conflicts during active motion have consequences that are best described by assuming that these internal model computations cease to influence neuronal responses, and can’t be described in linear terms. In this situation, introducing an ‘ad-hoc’ switching mechanism seems a sensible way to account for this experimental observation.

In fact, this feature of the model highlights that Cullen’s work has revealed two very different processes – which have never before been described in a single model – or even acknowledged as different. In (Roy and Cullen, 2004; Cullen et al., 2011; Cullen 2012; Carriot et al., 2013; Brooks and Cullen, 2014), the “sensory prediction mismatch” refers to neck proprioceptive errors. In (Brooks and Cullen, 2013, Brooks et al. 2015), “sensory prediction errors” refer to vestibular sensory errors. Cullen’s work has demonstrated that these two errors have fundamentally different roles in internal model computations. The former set of studies have clearly described the “switching” effect of proprioceptive mismatch; and the latter set has interpreted central vestibular responses as vestibular prediction errors (like the present study). However, the elementary difference between these mismatches has never been explained clearly in Cullen’s publications (in fact most of these publications use the term “sensory prediction” indiscriminately to refer to proprioceptive or vestibular predictions). We have now presented these points in the Introduction, and added a section addressing this in the Discussion.

In the second paragraph of the aforementioned subsection, the authors suggest that "proprioceptive mismatch" would cause to stop using the internal model. However, that seems a bit oversimplifying: Proprioceptive mismatch occurs due to a variety of natural conditions (e.g. increased head inertia due to carrying food, or due to fatigue, etc.) for which it would make no sense at all to completely stop using the internal model.

We apologize for this inadvertent oversimplification that would clearly be incorrect in other contexts than Cullen’s experiments, and we carefully reworded this section.

7) Parameter variations: Technically, systematic parameter variations to test for the robustness of the model predictions are missing (which can be easily addressed).

We re-designed Figure 10—figure supplement 2 to show the impact of varying (twofold increase or decrease) the parameters of the main model (canal and otoliths). We also tested variations of the parameter of the head and neck model but only described the results in the legend for brevity. Model predictions are generally very robust in the sense that changing model parameters won’t lead to numerical instability or affect simulation results qualitatively but, as expected, may affect the dynamic of the motion estimates.

8) One reviewer would also like to see more quantitative results showing time courses of adaptation in the model and e.g. firing rates match up with each other. This brings up the question of whether the model does adapt or whether all Kalman filters operate with steady state gain (as suggested by another reviewer).

Indeed, we did not model adaptation in this study. Although we are convinced that modeling the internal model’s adaptation would be very valuable work (in particular in connection with works on cerebellar learning as well as previous works on adaptation in the motor system, e.g. Berniker and Körding 2008), we thought that it would be best to keep it for a future study. The present manuscript is already packed with much information and a large number of main text and supplemental figures.